# Enhancing TCR specificity predictions by combined pan- and peptide-specific training, loss-scaling, and sequence similarity integration

**Mathias Fynbo Jensen, Morten Nielsen\***

Department of Health Technology, Section for Bioinformatics, Technical University of Denmark, Lyngby, Denmark

**\*For correspondence:**
morni@dtu.dk

**Competing interest:** The authors declare that no competing interests exist.

**Abstract** Predicting the interaction between Major Histocompatibility Complex (MHC) class I-presented peptides and T-cell receptors (TCR) holds significant implications for vaccine development, cancer treatment, and autoimmune disease therapies. However, limited paired-chain TCR data, skewed towards well-studied epitopes, hampers the development of pan-specific machine-learning (ML) models. Leveraging a larger peptide-TCR dataset, we explore various alterations to the ML architectures and training strategies to address data imbalance. This leads to an overall improved performance, particularly for peptides with scant TCR data. However, challenges persist for unseen peptides, especially those distant from training examples. We demonstrate that such ML models can be used to detect potential outliers, which when removed from training, leads to augmented performance. Integrating pan-specific and peptide-specific models alongside with similarity-based predictions, further improves the overall performance, especially when a low false positive rate is desirable. In the context of the IMMREP22 benchmark, this modeling framework attained state-of-the-art performance. Moreover, combining these strategies results in acceptable predictive accuracy for peptides characterized with as little as 15 positive TCRs. This observation places great promise on rapidly expanding the peptide covering of the current models for predicting TCR specificity. The NetTCR 2.2 model incorporating these advances is available on GitHub (https://github.com/mnielLab/NetTCR-2.2) and as a web server at https://services.health-tech.dtu.dk/services/NetTCR-2.2/.

## eLife assessment

This study presents a **useful** tool for predicting TCR specificity with **compelling** evidence for improvements over prior art. This work/tool will be broadly relevant to computational biologists and immunologists.

## Introduction

T-cell mediated immune responses play a crucial role in safeguarding the body's health by identification and elimination of pathogen infected and malfunctioning cells. One of the essential steps triggering the T-cell response is the recognition of peptides presented by MHC (Major Histocompatibility Complex) at the surface of cells by T cell receptors (TCR). The TCR is heterodimer (most often) formed by an α and a β chain. To be able to recognize the extreme variety of peptides presented by the MHC, the repertoire of different TCRs expressed by T cells in a given host is immense. This variation is mostly

limited to the interacting domains of the TCR, known as the complementary determining regions (CDRs) (*Davis and Bjorkman, 1988*).

The possibility of accurately predicting TCR specificity holds immense immunotherapeutic and biotechnological potentials, for instance as a means to rapidly and cost-effectively identify the target of relevant T cell populations in the context of antigen discovery, vaccine design and/or T cell therapy.

However, while machine-learning (ML) approaches have allowed to accurately predicts which peptides can be presented by the MHC (*Nielsen et al., 2020*), the scarce data, combined with the extreme variability of the TCR, has made it difficult to produce models with broad peptide-HLA coverage with similar accuracies for predictions of TCR specificity. Several models ranging from neural network models to similarity-based approaches have, however, allowed for development of accurate prediction models covering the limited set of peptides, for which sufficient data is available (*Hudson et al., 2023*).

Current ML-based methods for predicting TCR-specificity include convolutional neural network (CNN) models, such as ImRex (*Moris et al., 2021*), TCRAI (*Zhang et al., 2021*) and NetTCR 2.1 (*Montemurro et al., 2022*), auto-encoder-based models such as DeepTCR (*Sidhom et al., 2021*) decision-tree models such as SETE (*Tong et al., 2020*)**,** Gaussian process models such as TCRGP (*Jokinen et al., 2021*), as well as transformer-based models such as TULIP (*Meynard-Piganeau et al., 2023*). Furthermore, unsupervised similarity-based methods have been developed, such as TCRdist3 (*Mayer-Blackwell et al., 2021*), GLIPH2 (*Huang et al., 2020*) and TCRbase (*Montemurro et al., 2022*; *Shen et al., 2012*). However, many more models exist, and new models are constantly being proposed (refer to *Hudson et al., 2023* and *Meysman et al., 2023* for recent reviews).

Going back just a few years, the majority of models for TCR specificity predictions were based on single chain data, most often CDR3β, since this data was (and still is) much more abundant than paired chain data (e.g. comprising both the α- and β-chain). However, with the emergence of single-cell sequencing techniques, the volume of paired data has started to increase. Recent benchmarks have shown that training models on both chains leads to vastly improved predictive performance, compared to training on single chain data alone (*Meysman et al., 2023*). This performance is improved even further when also including the CDR1 and CDR2 sequences of the chains, either as amino acid sequences or implicitly through annotated V- and J-genes (from which the CDR1 and CDR2 sequences are determined).

While similarity-based methods have been shown to perform almost on par with ML-based models in cases where high similarity exist between the training and evaluation data and where many positive TCR observations are present for a given peptide, these approaches tend to be surpassed by ML methods when this similarity is decreased (*Montemurro et al., 2022*; *Meysman et al., 2023*).

Earlier work has been estimated that ~150 unique TCRs are required to construct an accurate ML prediction model capturing the rules of TCR specificity towards a specific peptide (*Montemurro et al., 2021*), and that very limited if any predictive power can be maintained when predicting specificity towards peptides, for which no binding TCRs have been recorded (*Moris et al., 2021*; *Grazioli et al., 2022*). This lack of extrapolative power is the single most current challenging factor within the field of TCR specificity prediction. In order to predict binding for unseen peptides, models are required to be trained in a pan-specific setup, where a model is trained on data covering many different peptides at once including the peptide sequence as input to the model. Such a setup has with high success been applied for the MHC system where pan-specific models have been developed on data spanning large sets of different MHC molecules resulting in high extrapolation power also for molecules not included in the training data (*Reynisson et al., 2020*; *Nilsson et al., 2023*).

While many of the current day models for TCR specificity predictions are trained in this way, no models have so far been able to obtain substantial performance when predicting binding for unseen peptides that are not highly similar to already seen peptides. The main problem limiting the power of extrapolation for these pan-specific models lies in the scarcity of data available for training, especially so for paired-chain data, combined with the problem that the current data is highly imbalanced towards only a few peptides. Moreover, while the availability of data has increased recently, another problem is the high proportion of noise contained within the data produced with the current single-cell high-throughput sequencing methods (*Zhang et al., 2021*; *Montemurro et al., 2023*). Statistical denoising methods have been proposed to deal with this problem (*Zhang et al., 2021*; *Povlsen et al., 2023*). However, these methods are naturally challenged when dealing with small T cell populations,

and due to their statistical nature likely share suboptimal sensitivity (i.e. remove true data) and specificity (i.e. allow false positives to slip through) (*Zhang et al., 2021*; *Montemurro et al., 2023*).

In this manuscript, we seek to address these issues in the context of a large data set of paired TCRs with annotated pMHC specificity. We investigate impacts of refining the machine learning model architecture and training setup to achieve pan-specific models with improved generalization capabilities. Further, strategies such as data denoising in terms of outlier identification in the training data, and inclusion of redundant data during training, is explored. We also investigate a new model architecture which combines the properties of a pan- and peptide-specific model, and explore how a similarity based approach can be integrated into the framework to boost model performance.

## Materials and methods
### Training data
The initial data was acquired from IEDB (*Vita et al., 2019*) and VDJdb (*Bagaev et al., 2020*) on the 23rd and 24th of August 2022, respectively, using a query to select only positive T-cell assays for MHC class I and Human cells. Additionally, only paired-chain (αβ) data was collected. This resulted in a dataset of 21,825 observations across 631 peptides for IEDB and 27,005 observations across 898 peptides for VDJdb.

This data was subsequently filtered to exclude data originating from 10 X sequencing, which was done by manually investigating references with at least 100 observations. Furthermore, filtering was conducted to include only observations with annotated V and J genes and fully specified MHC alleles. In cases where the V and J genes did not have a fully specified allele, the most common allele (*01) was assigned. Furthermore, CDR3 sequences which did not follow the nomenclature of beginning with a cysteine and ending with a phenylalanine (F) or tryptophan (W) were modified to follow this nomenclature by adding a cysteine to the start of the sequence if missing, and adding phenylalanine to the end of the sequence if phenylalanine or tryptophan was not present at the end of the sequence. This filtering resulted in 4439 observations across 405 peptides after merging the two datasets together and dropping duplicate entries.

Next, a dataset from a 10 x sequencing study (*10x Genomics, 2020*) which was denoised with iTRAP (*Povlsen et al., 2023*) was included, resulting in a combined dataset of 10,239 observations across 435 peptides.

To retrieve the full TCR sequences required for annotating all CDRs, *Stitchr* (*Heather et al., 2022*) was used. In brief, *Stitchr* looks up the sequences for the V and J genes in IMGT/GENE-DB and attempts to align these sequences with the specified CDR3 amino acid sequence. In case of mismatches in the alignment, the CDR3-proximal residues of the V and J gene products, respectively, are progressively removed until a match can be found. As the alignment failed on either one chain or both for some of the sequences, 9045 full TCR sequences were retrieved in this step.

In cases where *Stitchr* failed to reconstruct the TCR, a second run of *Stitchr* was performed where tryptophan was added instead of phenylalanine for the CDR3s with the wrong nomenclature. This resulted in the rescue of 20 additional TCR sequences, bringing the total number of full TCR sequences up to 9065 (88.5% of the inputs given to *Stitchr*).

Finally, the CDR1, CDR2, and CDR3 amino acid sequences were annotated by submitting the full TCR sequences to the *ANARCI* software (*Dunbar and Deane, 2016*), which is a tool that is used for annotating the sequences according to the IMGT naming scheme (*Lefranc et al., 2003*). Here, CDR1 was defined as position 27–38, CDR2 as position 56–65 and CDR3 as position 105–117.

### Redundancy reduction
The CDR-annotated data was redundancy reduced in two steps using the Hobohm 1 algorithm (*Hobohm et al., 1992*) based on a summed BLOSUM62 encoded kernel similarity (*Shen et al., 2012*) of CDR3α and CDR3β. In the first step, the dataset was split according to peptides, and a redundancy reduction was carried out separately for TCRs belonging to each unique peptide using a 0.95 kernel similarity threshold. Here, only peptides with at least 30 unique TCRs after the first redundancy reduction were kept. This redundancy reduction and filtering resulted in a dataset of 6415 observations across 26 peptides.

**Table 1.** Per peptide overview of the full positive training data.

The source organism for each epitope, as well as the MHC allele which they bind to, are here shown. Additionally, the number of observations discarded during each redundancy reduction step, as well as the total remaining number of observations, are also listed, along with the number of observations originating from 10 x sequencing.

| Peptide | Organism | MHC | Pre reduction count | Removed in first reduction | Removed in second reduction | Post reduction count | Not 10 X | 10 X |
|---|---|---|---|---|---|---|---|---|
| GILGFVFTL | Influenza A virus | HLA-A*02:01 | 1897 | 645 | 127 | 1125 | 426 | 699 |
| RAKFKQLL | Epstein Barr virus | HLA-B*08:01 | 1065 | 114 | 17 | 934 | 0 | 934 |
| KLGGALQAK | Human CMV | HLA-A*03:01 | 912 | 8 | 2 | 902 | 0 | 902 |
| AVFDRKSDAK | Epstein Barr virus | HLA-A*11:01 | 725 | 5 | 4 | 716 | 0 | 716 |
| ELAGIGILTV | Melanoma neoantigen | HLA-A*02:01 | 435 | 6 | 3 | 426 | 55 | 371 |
| NLVPMVATV | Human CMV | HLA-A*02:01 | 384 | 43 | 11 | 330 | 154 | 176 |
| IVTDFSVIK | Epstein Barr virus | HLA-A*11:01 | 323 | 13 | 2 | 308 | 0 | 308 |
| LLWNGPMAV | Yellow fever virus | HLA-A*02:01 | 322 | 72 | 21 | 229 | 229 | 0 |
| CINGVCWTV | Hepatitis C virus | HLA-A*02:01 | 231 | 4 | 1 | 226 | 75 | 151 |
| GLCTLVAML | Epstein Barr virus | HLA-A*02:01 | 278 | 59 | 7 | 212 | 95 | 117 |
| SPRWYFYYL | SARS-CoV2 | HLA-B*07:02 | 158 | 4 | 5 | 149 | 149 | 0 |
| ATDALMTGF | Hepatitis C virus | HLA-A*01:01 | 128 | 21 | 4 | 103 | 0 | 103 |
| DATYQRTRALVR | Influenza A virus | HLA-A*68:01 | 100 | 4 | 3 | 93 | 93 | 0 |
| KSKRTPMGF | Hepatitis C virus | HLA-B*57:01 | 115 | 14 | 12 | 89 | 0 | 89 |
| YLQPRTFLL | SARS-CoV2 | HLA-A*02:01 | 69 | 6 | 1 | 62 | 54 | 8 |
| HPVTKYIM | Hepatitis C virus | HLA-B*08:01 | 60 | 5 | 2 | 53 | 0 | 53 |
| RFPLTFGWCF | HIV-1 | HLA-A*24:02 | 58 | 7 | 0 | 51 | 51 | 0 |
| GPRLGVRAT | Hepatitis C virus | HLA-B*07:02 | 51 | 3 | 0 | 48 | 0 | 48 |
| CTELKLSDY | Influenza A virus | HLA-A*01:01 | 48 | 0 | 0 | 48 | 48 | 0 |
| RLRAEAQVK | Epstein Barr virus | HLA-A*03:01 | 47 | 0 | 0 | 47 | 0 | 47 |
| RLPGVLPRA | AML neoantigen | HLA-A*02:01 | 43 | 0 | 0 | 43 | 0 | 43 |
| SLFNTVATLY | HIV-1 | HLA-A*02:01 | 38 | 0 | 0 | 38 | 0 | 38 |
| RPPIFIRRL | Epstein Barr virus | HLA-B*07:02 | 40 | 2 | 2 | 36 | 24 | 12 |
| FEDLRLLSF | Influenza A virus | HLA-B*37:01 | 31 | 0 | 0 | 31 | 31 | 0 |
| VLFGLGFAI | T1D neoantigen | HLA-A*02:01 | 32 | 1 | 0 | 31 | 31 | 0 |
| FEDLRVLSF | Influenza A virus | HLA-B*37:01 | 36 | 0 | 13 | 23 | 23 | 0 |

A second redundancy reduction was subsequently carried out also at a 0.95 kernel similarity threshold across all remaining observations and peptides, where the data was sorted by peptide according to TCR count (least abundant to most abundant) in order to limit the risk of removing observations from peptides with few observations. This resulted in the further removal of 68 observations, resulting in a final dataset of 6,353 positive observations across 26 peptides. The amount of redundant data removed by the redundancy reductions are summarized in *Table 1*, as well as information regarding source organism and MHC allele for each peptide, and number of observations originating from 10 X sequencing data. The vast majority of 10 X data comes from the iTRAP filtered dataset, with a few observations originating from other 10 X studies that managed to slip through the initial manual filtering.

## Data partitioning and generation of swapped negatives

To prepare the data for model training, this data was randomly split into five partitions, and negatives were generated by swapping the TCRs for a given peptide with TCRs binding to other peptides.

Here, such TCRs were only samples from peptides which had a Levenshtein distance greater than 3, to reduce the risk of generating false negatives. For each positive observation, five negative observations were generated using this approach, except for the GILGFVFTL peptide, where all TCRs from the other peptides were used as negatives, since there was not enough data to allow for a 1:5 positive to negative ratio for this peptide (a 1:4.647 ratio was achieved here). The generation of swapped negatives was done separately within each partition, in order to reduce the risk of data leakage.

## Baseline model

TCRbase (*Montemurro et al., 2022*), a distance-based model, was used as the baseline model. For a given peptide, TCRbase calculated the similarity between sets of CDRs found in the test partition to all positive CDR sets found in the remaining partitions. In short, the similarity is calculated per CDR as the mean kernel-similarity of BLOSUM62-encoded kmers ranging from size 1–30 between the two sets of CDRs that are compared (*Shen et al., 2012*). The weighting for the CDRs was set to 1,1,3,1,1,3 for CDR1α-, CDR2α-, CDR3α-, CDR1β-, CDR2β-, and CDR3β, respectively, in line with earlier recommendations (*Montemurro et al., 2022*).

## CNN architecture

The CNN architecture for NetTCR 2.1 (*Montemurro et al., 2022*) was reconstructed in Keras (*Chollet, 2015*), in preparation for further updates to the architecture. In brief, the original architecture consists of a set of convolutional 1D layers for each input feature, where each layer has 16 filters of kernel size of 1, 3, 5, 7, and 9, respectively, which are activated by a sigmoid activation function. Each layer is then max-pooled, concatenated, and fed to a dense layer of size 32 followed by a linear output layer of size 1, representing the final prediction score. The outputs of both linear layers are activated by a sigmoid activation function.

Except for the first models referred to as NetTCR 2.1 (which ran in PyTorch *Paszke et al., 2019*), the version 2.2 CNN models described in this paper used a slightly modified architecture compared to NetTCR 2.1. Here, the activation function for the max-pooling layer was replaced with a rectified linear unit (previously sigmoid), a dropout layer was introduced for the concatenated max-pooling output, and the size of the dense layer was doubled to 64 neurons. For the models utilizing dropout, a dropout rate of 0.6 was used. The models referred to here as NetTCR 2.1 uses the original pan-specific NetTCR 2.1 architecture (*Montemurro et al., 2022*), which also includes convolutional filters for the peptide-sequence.

## Embedding

The input features for the CNN models consisted of peptide-, CDR1α-, CDR2α-, CDR3α-, CDR1β-, CDR2β-, and CDR3β-amino acid sequence. These were each represented using a BLOSUM50-embedding (calculated using a normalization factor of 5) and right-padded to the maximum length observed for that feature in the dataset, by assigning a vector of 20 times –1 for each missing residue. For reference, the maximum length observed was 12, 7, 8, 22, 6, 7, and 23 residues for the peptide-, CDR1α-, CDR2α-, CDR3α-, CDR1β-, CDR2β-, and CDR3β-amino acid sequences, respectively.

## Training setup and early stopping

All CNN models were trained in a nested cross-validation setup with four folds in the inner loop and fivefolds in the outer loop. Here, three partitions were used for training, one was used for validation, while the remaining partition was used as a test partition to evaluate the performance of the model. For all CNN models, Binary Cross Entropy was used as the loss function, and the Adam optimizer (*Kingma and Ba, 2014*) was used for updating the weights during training. A learning rate of 0.001 was used for training of all models.

A patience of 200 epochs was used for the early stopping for the peptide-specific CNNs, whereas for the pan-specific CNNs, a patience of 100 epochs was used. The increased patience for the peptide-specific models was introduced to allow the models to escape local minima imposed by small training set sizes. For the NetTCR 2.1 models (PyTorch) (*Paszke et al., 2019*), the validation loss was used as a stopping criterion for early stopping, and validation AUC 0.1 was used as the stopping criterion for the updated models in Keras.

For the pan-specific models, a batch size of 64 was used together with shuffling. For the peptide-specific models, an adaptive batch size was used, which ensured that no batch ended up having less than 32 observations. Here, it was first tested if it was possible to use a batch size of 64 while still having at least 32 observations for the final batch. If not, the default batch size of 64 was progressively increased by 1, until it was ensured that the final batch had at least 32 observations.

## Performance evaluation

The cross-validation setup results in four models generated in the inner loop. The test set predictions were then calculated from the average over the four predictions for each entry. The performance was evaluated on the five concatenated test sets in terms of AUC and AUC 0.1 on a per-peptide basis, as well as the unweighted and weighted average performance across all peptides:

$$M_{unweighted} = \frac{\sum_{peptide} M_{peptide}}{N_{unique\ peptides}}$$

$$M_{weighted} = \sum_{peptide} M_{peptide} \cdot \frac{N_{peptide}}{N_{total}}$$

where $M_{unweighted}$ and $M_{weighted}$ is the unweighted and weighted average metric, respectively, $M_{peptide}$ is the metric for a given peptide, $N_{peptide}$ is the number of positive observations for a given peptide, $N_{unique\ peptides}$ is the number of unique peptides, and $N_{total}$ is the total number of positive observations across all peptides.

A summary of the per-peptide performance of all models is found in *Supplementary file 1*.

## Performance comparisons

To assess the difference in performance between models, bootstraps were performed by sampling with replacement from the model predictions 10,000 times and calculating the weighted and unweighted performance metrics for each subsample as described above. The same seed for subsampling and order of predictions was used for all bootstraps, to ensure that performance within a given subsample could be compared between models. The p-value for the null hypothesis that two models had equal performance was then calculated as the number of times that the first model had a higher performance than the second model within the same subsample, normalized by the total number of subsamples.

## Weighted loss

A weighted loss was implemented for the pan-specific CNN model to allow the model to focus more on the observations from the less abundant peptides in the training dataset. Here, the binary cross entropy loss for observations from each peptide was weighted according to the formula:

$$loss_{weighted} = \frac{log2\left(\frac{N_{total}}{N_{peptide}}\right)}{c}$$

where $N_{total}$ is the total number of observations, $N_{peptide}$ is the number of observations for the given peptide, and $c$ is a constant that is used to scale the loss, so the overall loss becomes close to that of the unweighted approach. The value of $c$ was set to 3.8 to ensure that the overall weighted loss was comparable to the training done without sample weighting. For the peptide-specific models, a weight of 1 was used for all samples.

## Redundant training dataset

A dataset was constructed based on the primary training dataset, where redundant data from the first redundancy reduction (see *Table 1*) was added back by assigning them to the partition of the data point that they were redundant to. Only positive data was added back in this way, and additional swapped negatives were not generated for this dataset to keep it as similar to the original as possible. Models trained on this dataset were evaluated on the original test datasets without redundant data.

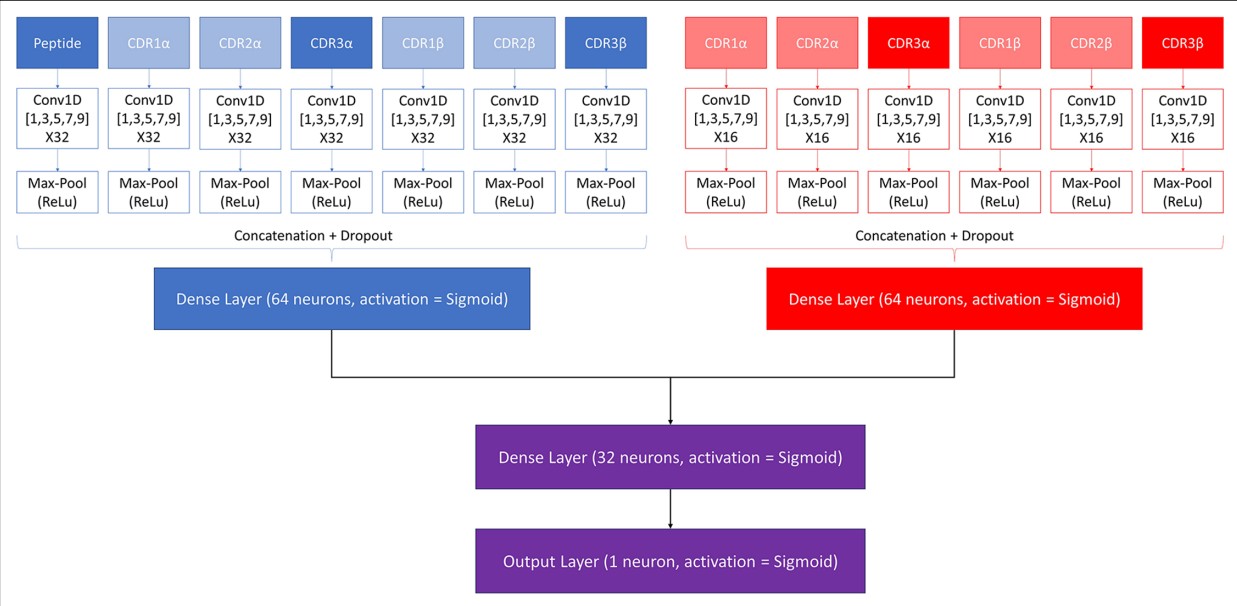

**Figure 1.** Architecture of the pre-trained model. The pan-specific CNN block consists of the layers shown in blue, whereas the peptide-specific CNN block consists of the layers shown in red. During the pan-specific training, the weights and biases for the peptide-specific CNN block are frozen, whereas the opposite is the case during the peptide-specific training. The layers shown in purple are kept unfrozen during both training steps.

## Limited training dataset

Using the prediction scores for the validation partitions of the updated peptide-specific CNN model, additional datasets were constructed by removing observations that consistently received a poor prediction score in relation to their designated label. That is, positive observations were removed if they received a validation prediction score of less than the nth percentile of the negative prediction scores for the given peptide for all four models that were not trained on that partition, while negative observations were removed if they received a validation prediction score of more than the $(1 - n)$th percentile of the positive validation prediction scores for all four models that were not trained on that partition. Thresholds of n=50, 60, 70, 80, 85, 90, and 95 were tested in this way.

## Pre-trained models

A modified version of the NetTCR 2.2 architecture was made to combine the properties of the pan- and peptide-specific models, as shown in *Figure 1*. This architecture consists of a pan-specific and a peptide-specific CNN block. The pan-specific CNN block consists of 32 1D convolutional filters of size 1, 3, 5, 7, and 9, respectively for each of the peptide-, CDR1α-, CDR2α-, CDR3α-, CDR1β-, CDR2β-, and CDR3β embeddings. The peptide-specific CNN block consists of 16 1D convolutional filters, also of size 1, 3, 5, 7, and 9, respectively, for the same feature embeddings, except the peptide embedding, as this information is redundant when trained on a single peptide. The outputs from each CNN block are max-pooled with a rectified linear unit activation function, concatenated, and fed to two dropout layers with a dropout rate of 0.6, one for each output of a CNN block.

Each of the two resulting tensors are fed separately to dense layers with 64 units and sigmoid activation, both of which are connected to a second dense layer with 32 units and a sigmoid activation. The output of the second dense layer is finally connected to an output layer of size 1, which is also activated by a sigmoid activation function, to give a prediction score between 0 and 1.

These models are trained in two rounds. During the first round of training, a pan-specific training is performed. Here the weights in the peptide-specific CNN block are kept frozen, as shown in *Figure 1*. This pre-trained model is then used as the starting point for a second round of training performed in a peptide-specific setup, where the weights in the pan-specific CNN block are frozen, while those in the peptide-specific CNN block are unfrozen. During both training rounds, a patience of 100 is used and the maximum number of epochs is set to 200.

## CNN – TCRbase ensemble

The pre-trained CNN model was combined with the sequence similarity based TCRbase model (*Montemurro et al., 2022*; *Shen et al., 2012*). The predictions for this new ensemble were calculated using the following formula:

$$P_{TCRbase\ ensemble} = P_{CNN} \cdot P_{TCRbase}^{\alpha}$$

where $P_{TCRbase\ ensemble}$ is the prediction of the combined ensemble, $P_{CNN}$ is the prediction of the CNN model, $P_{TCRbase}$ is the prediction from TCRbase, and α is a scaling factor used to give TCRs with low similarity to known binders a harsher penalty. This ensemble was tested on the validation partitions of the full dataset, where α was varied from 0 to 40.

The Pearson correlation coefficients between the α resulting in the best performance in terms of AUC and AUC 0.1, respectively, and the corresponding performance metric for the TCRbase and pre-trained model without scaling, was calculated using the *pearsonr* function from *scipy.stats* (*Virtanen et al., 2020*). Five samples were used for each peptide, as there were five different validation partitions to consider, resulting in a total of 130 samples for calculating the Pearson correlation coefficients. p-Values for the null hypothesis that there was no correlation was also reported using this function.

## Percentile-rank rescaling

Prediction scores were rescaled to a percentile rank by comparing the score to the score distribution obtained for 15,957 negative controls paired to the corresponding peptide. These negative controls were obtained from the IMMREP 2022 workshop dataset (*Meysman et al., 2023*). Here, the percentile rank score for a given TCRs was calculated as the percentage of negative controls which had a score above the score of that of the TCR.

## Peptide specificity test

To evaluate the models' ability to correctly identify which peptide is most likely to bind a given TCR, all TCRs were paired with all peptides present within each partition, and predictions were performed by the models which had not seen the given partition during training. The specificity was then calculated per peptide as the number of times that the true peptide-TCR complex was given the highest prediction score, compared to the total number of positive observations in the original dataset for the given peptide. The test was performed on the limited dataset, where the peptides KLGGALQAK, AVFDRKSDAK, NLVPMVATV, CTELKLSDY, RLRAEAQVK, RLPGVLPRA, and SLFNTVATLY were discarded, due to low performance of the full model (AUC 0.1<0.65).

## Leave most out

To test the models' ability to learn from small data sets, models were re-trained on small subsets of the original data. For each of the peptides with at least 100 positive observations in the limited training dataset except for KLGGALQAK, AVFDRKSDAK, and NLVPMVATV (e.g. GILGFVFTL, RAKFKQLL, ELAGIGILTV, IVTDFSVIK, LLWNGPMAV, CINGVCWTV, GLCTLVAML and SPRWYFYYL were included), new training datasets were constructed by subsampling 5, 10, 15, 20, 25, 50, and 100 positive peptides, respectively, per partition, as well as five negative observations per positive. KLGGALQAK, AVFDRKSDAK, and NLVPMVATV were excluded from this analysis, due to low performance of the full model (AUC 0.1<0.65). All models trained here were evaluated on the full dataset (not limited).

As a baseline, TCRbase was used to perform predictions on the test partitions, using the positives from the four remaining partitions as the positive database for similarity inference.

In addition, a set of peptide-specific models were also trained on these datasets, using the same hyperparameters as the best (non-pre-trained) peptide-specific model, when evaluated on the full dataset.

A set of pre-trained models were also re-trained on these datasets, where the first training round of the pan-specific CNN was conducted on the leave one out dataset. For each peptide and each number of positives, the pan-specific CNN block was fine-tuned by training for 30 epochs in a pan-specific setup, where observation for the leave-most-out peptide was assigned a sample weight of 1, while the observations for the remaining peptides were assigned a weight of 0.1. Swapped negatives assigned to other peptides than the one the models were trained for were removed for this training, if they originated from an observation belonging to the peptide in question. Following this,

**Table 2.** Overview of number of TCRs for each peptide in the IMMREP 2022 training dataset before and after redundancy reduction. The redundancy reduction was performed using a kernel similarity threshold of 95%.

| Peptide | Pre reduction count | Post reduction count | Percent redundant |
|---|---|---|---|
| All | 2445 | 1960 | 19.8% |
| GILGFVFTL | 544 | 301 | 44.7% |
| NLVPMVATV | 274 | 242 | 11.7% |
| YLQPRTFLL | 267 | 227 | 15.0% |
| TTDPSFLGRY | 193 | 187 | 3.1% |
| LLWNGPMAV | 188 | 175 | 6.9% |
| CINGVCWTV | 183 | 179 | 2.2% |
| GLCTLVAML | 146 | 91 | 37.7% |
| ATDALMTGF | 104 | 78 | 25.0% |
| LTDEMIAQY | 100 | 94 | 6.0% |
| SPRWYFYYL | 92 | 92 | 0.0% |
| KSKRTPMGF | 85 | 63 | 25.9% |
| NQKLIANQF | 56 | 53 | 5.4% |
| HPVTKYIM | 48 | 41 | 14.6% |
| TPRVTGGGAM | 45 | 44 | 2.2% |
| NYNYLYRLF | 44 | 42 | 4.6% |
| GPRLGVRAT | 40 | 37 | 7.5% |
| RAQAPPPSW | 36 | 14 | 61.1% |

the pan-specific CNN block was frozen, and the peptide-specific CNN block was trained on the observations for the peptide of interest.

Finally, an ensemble consisting of the pre-trained models scaled by the TCRbase prediction ($\alpha$=10) were evaluated (see CNN - TCRbase ensemble).

Due to the low number of positives for some of the leave-most-out datasets, the default batch size was set to 32 for the peptide-specific training, while the criteria for early stopping and model saving was changed from validation AUC 0.1 to a custom metric taking both validation AUC 0.1 and binary cross entropy loss into account. This custom metric was calculated as:

$$CM_{val} = AUC\,0.1_{val} - Loss_{val} \cdot 0.1$$

and the model was saved when this value was maximized. A patience of 100 was used for early stopping during the peptide-specific training.

## IMMREP 2022 training and evaluation

The labeled training and test data for the IMMREP 2022 workshop (*Meysman et al., 2023*) was collected from GitHub (GitHub - viragbioinfo/IMMREP_2022_TCRSpecificity; *viragbioinfo et al., 2022*) on the 5th of July 2023. The training data was randomly split into five partitions, and models were trained in the same cross-validation as described above, for example nested cross-validation for the neural network models and a fivefold cross-validation for TCRbase. To make the data compatible with our models, the labels for the negative observations were changed from –1 to 0. The performance of each model was then evaluated on the separate test dataset, using the average in prediction score given by all models resulting from the cross-validation.

A separate redundancy reduced dataset was created based on the IMMREP dataset following the strategy described above. An overview of the number of observations removed by this redundancy reduction is shown in *Table 2*.

Swapped negatives were generated within each partition, by randomly sampling TCRs binding to other peptides with a Levenshtein distance of at least three, until a 1:3 ratio of positives to negatives were achieved. Negative controls were first subjected to a redundancy reduction at a 95% similarity threshold, followed by random partitioning. Within each partition, negative controls were sampled in a 1:2 ratio of positive to negatives for each peptide, bringing the total positive to negative ratio up to 1:5.

Models were then trained on this training dataset using nested cross-validation (or fivefold cross-validation for TCRbase), while the performance was evaluated on the test-partitions, which were not seen during training. The average prediction score of the four cross-validation models per test partition was used as the final prediction score for this performance evaluation. A summary of the per-peptide performance of all models trained and tested on the IMMREP 2022 dataset is found in **Supplementary file 2**.

## Results

Here, we seek to demonstrate step by step how improved low complexity models with state-of-the-art performance for the prediction of TCR specificity can be obtained by dealing with the essential issues related to data imbalance, low data accuracy and data volume. We do this on a large set of data obtained from the public domain covering paired full length TCR sequences with specificity annotated towards a set of 26 unique peptides (for details refer to Materials and methods). The machine learning framework applied is a low complexity max-pooled CNN architecture inspired by the original NetTCR model (**Montemurro et al., 2021**). This model makes use of 80 convolutional filters for the peptide and each of the 6 CDRs. Due to the limited number of peptides (and HLAs), HLA is not included in the model.

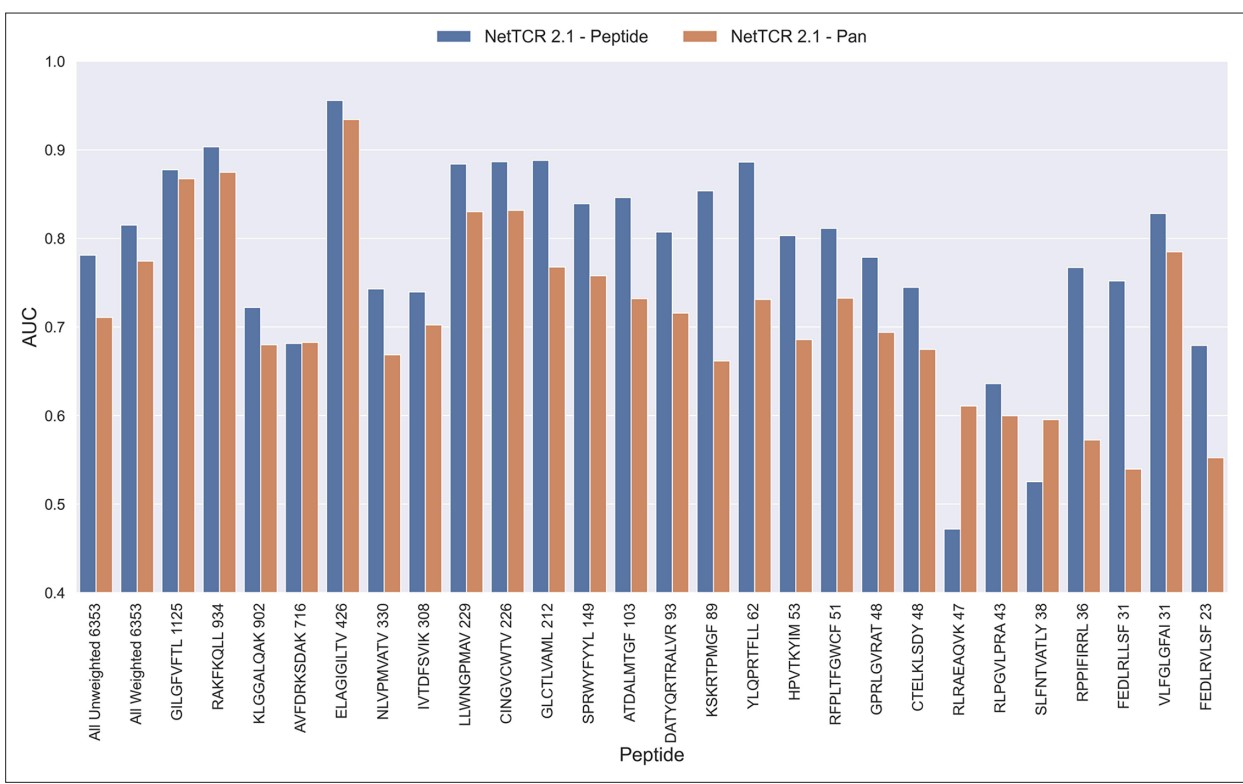

**Figure 2.** Per peptide performance of the peptide-specific and pan-specific NetTCR 2.1 in terms of AUC, when trained and evaluated on the new dataset. The peptides are sorted based on the number of positive observations from most abundant to least abundant, with the number of positive observations listed next to the peptide sequence. The unweighted (direct) mean of AUC across all peptides is shown furthest to the left, while the weighted mean is shown second furthest to the left. The weighted mean is weighted by the number of positive observations per peptide and puts more emphasis on the peptides with the most observations. The models included in this figure corresponds to model 1 (NetTCR 2.1 - Pan) and model 2 (NetTCR 2.1 - Peptide) in **Supplementary file 1**.

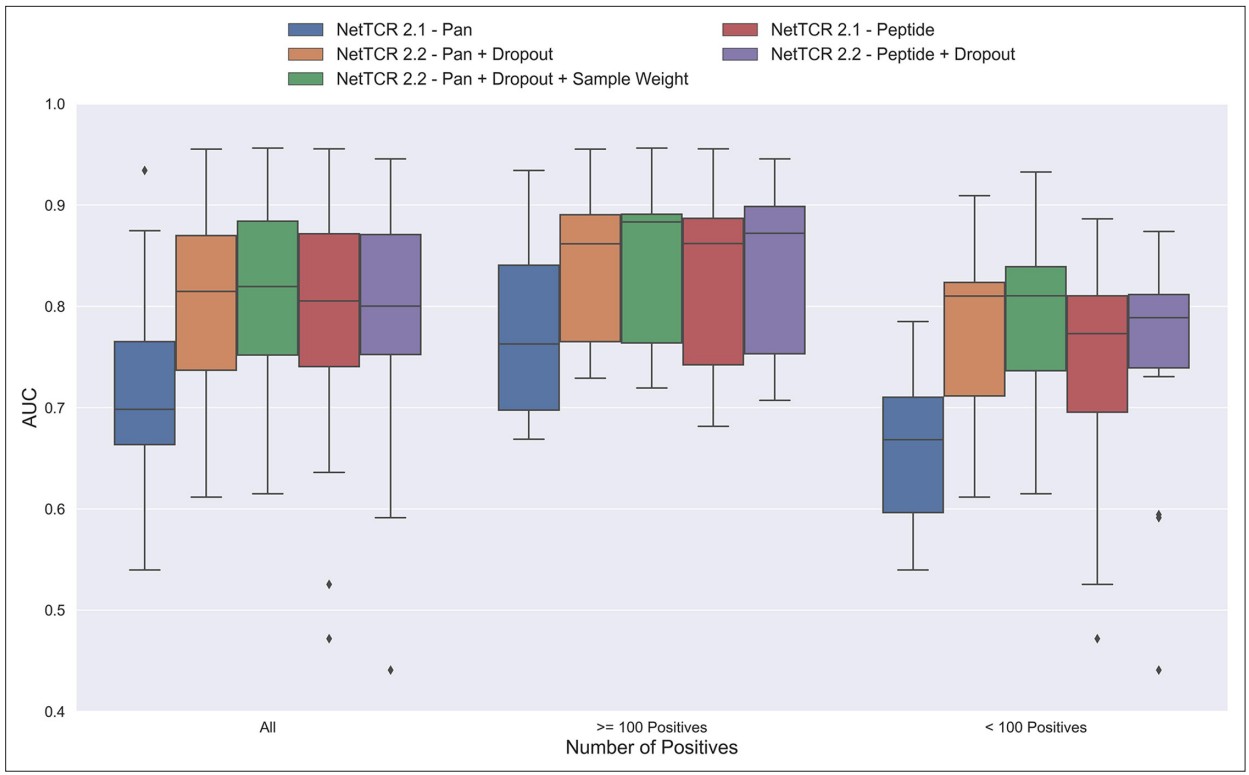

**Figure 3.** Boxplot of AUC of the pan- and peptide-specific NetTCR 2.1 and 2.2 models, respectively. The NetTCR 2.2 models include the updates to the model architecture, with the primary change being the introduction of dropout for the concatenated max-pooling layer (dropout rate = 0.6). Both the introduction of dropout and sample weights are shown to result in considerably improved performance for the pan-specific model. Separate boxplots are shown for all peptides, as well as separately for peptides with at least 100 positive observations and peptides with less than 100 positive observations, to highlight the effect of introducing dropout and sample weight for the least abundant peptides. The models included in this figure corresponds to model 1 (NetTCR 2.1 - Pan), model 3 (NetTCR 2.2 - Pan +Dropout), model 4 (NetTCR 2.2 - Pan +Dropout + Sample Weight), model 2 (NetTCR 2.1 - Peptide) and model 5 (NetTCR 2.2 - Peptide +Dropout) in *Supplementary file 1*.

The online version of this article includes the following figure supplement(s) for figure 3:

**Figure supplement 1.** Boxplot of AUC 0.1 of the pan- and peptide-specific NetTCR 2.1 and 2.2 models, respectively.

The NetTCR framework has so far performed best in a peptide-specific setup where separate models are trained for individual peptides (*Montemurro et al., 2022*). Ideally, one would like to construct pan-specific models trained across multiple peptides at once, since this should allow the model to leverage shared information resulting in boosted predictive power, especially for peptides characterized with few or even no positive TCR observations. However, for NetTCR 2.1, the opposite tendency was observed. This work was however limited to only 6 peptides, and we therefore first investigated if this conclusion still held true in the context of our data set with increased peptide coverage. The result of this analysis can be seen in *Figure 2* and demonstrates that peptide-specific models also here are superior to the pan-specific model.

## Improving the pan-specific model
### Updating the model architecture for pan-specific predictions
One potential source of the low performance for the pan-specific model is the high imbalance in the number of observations per peptide resulting in the model focusing/overfitting on the more abundant peptides. To investigate this, we first introduced a dropout-layer with a dropout rate of 0.6 to the architecture for the concatenated output of the max-pooling layer, while also doubling the number of neurons for the dense layer from 32 to 64 to allow for sufficient flow of information. Additionally, this model was rebuilt in Keras (*Chollet, 2015*) and the stopping criterion was changed from validation loss to validation AUC 0.1. As shown in *Figure 3*, this resulted in a highly significant increase in performance (bootstrap test resulting in p<0.0001 for all tested metrics).

To further deal with the imbalance problem, we next introduced a peptide specific sample weight so that the loss was increased for peptides with a low number of positive observations (for details refer to Materials and methods). This is based on the notation that the model then would focus more on the less abundant peptides when updating the weights. As demonstrated in *Figure 3*, this approach resulted in a further increase in performance for the less abundant peptides, whereas the performance for the more abundant peptides was largely unaffected. Here, a significant increase in performance was observed for the unweighted mean AUC (p=0.0026) and AUC 0.1 (p<0.0001).

Moreover, when only considering the peptides with less than 100 positive observations, the improvement in performance was significant across all metrics (p=0.0101, p=0.0035, p<0.0001 and p<0.0001 for AUC, weighted AUC, AUC 0.1 and weighted AUC 0.1, respectively).

Next, the impacts of the updates to the model architecture and training strategy on the performance of peptide-specific models was investigated. As expected, these results (*Figure 3* and *Figure 3—figure supplement 1*) demonstrated a limited gain in performance compared to NetTCR-2.1 - Peptide, which was however significant for all metrics (p=0.0337 for AUC, and p<0.0001 for AUC 0.1 and weighted AUC/AUC 0.1). Interestingly, the updated pan-specific model significantly outperformed the updated peptide-specific models in terms of both unweighted (p<0.0001) and weighted AUC (p=0.0004), and the performance gain was especially observed for the less abundant peptides. However, in terms of AUC 0.1, the updated peptide-specific model (NetTCR-2.2 - Peptide) maintained a superior performance (see *Figure 3—figure supplement 1*) (p=0.0008 and p<0.0001 for AUC 0.1 and weighted AUC 0.1, respectively). We will later address how to get the best of the two models later in the Pre-training section.

## Reusing redundant data does not lead to better performance
The results until now have been generated based on redundancy reduced data. That is data where redundant data have been removed based on a Hobohm-1 like redundancy reduction algorithm (for

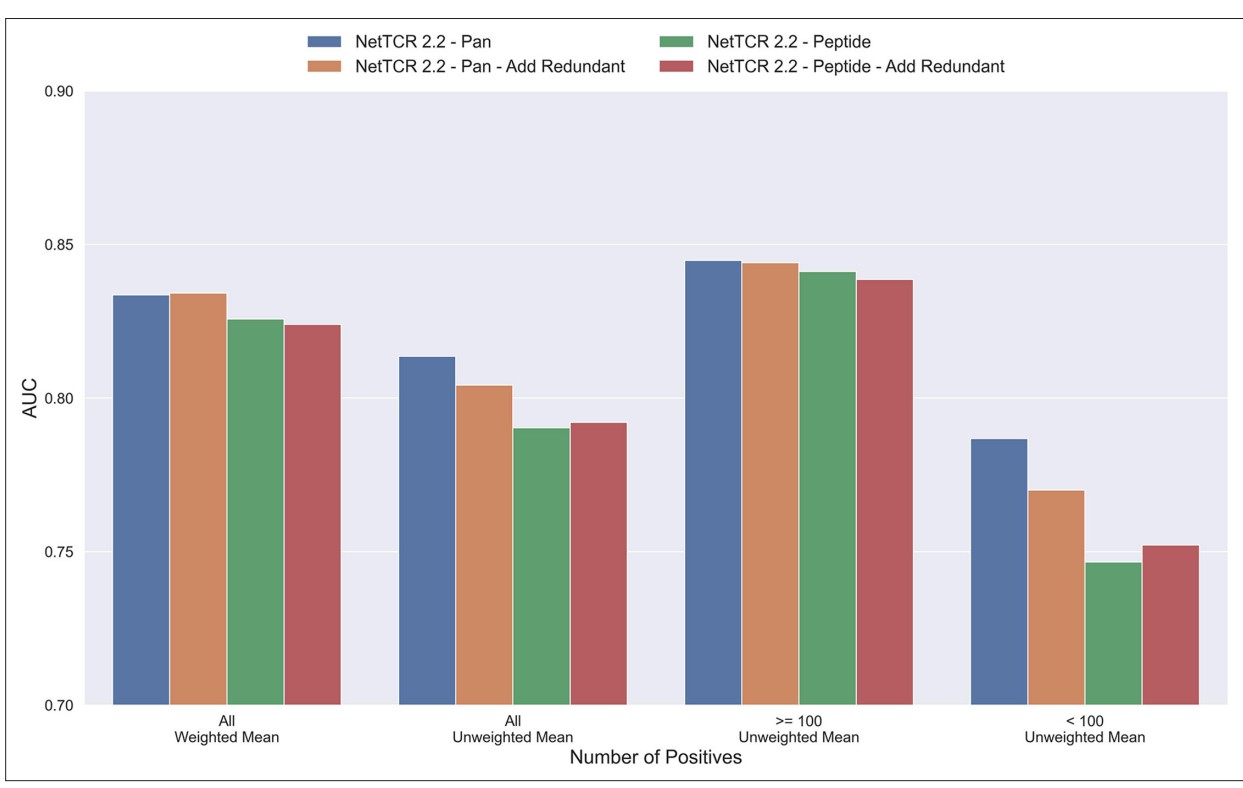

**Figure 4.** Mean AUC of the pan-specific and peptide-specific NetTCR 2.2 models, when training on the original redundancy reduced training data, and with redundant observations back. The AUC is reported in terms of weighted and unweighted mean across all peptides, as well as unweighted mean when the data is split into peptides with at least 100 positive observations, and less than 100 positive observations. The models included in this figure corresponds to model 4 (NetTCR 2.2 - Pan), model 6 (NetTCR 2.2 - Pan - Add Redundant), model 5 (NetTCR 2.2 - Peptide), and model 7 (NetTCR 2.2 - Peptide - Add Redundant) in *Supplementary file 1*.

details see Materials and methods). However, as data is very sparse, one could argue that a better approach would be to reuse redundant data, either by performing clustering when making the data partitions, or by adding back redundant data to the same partition as the data that it was redundant to. To test how such a strategy would affect the performance of the model, a new dataset was created using the latter approach. To keep the performance evaluation fair, redundant data were only re-introduced to the training dataset while the original dataset without redundant observations was used for testing and performance evaluation. The total number of redundant observations for each peptide from the first redundancy reduction is shown in *Table 1* (note that those from the second reduction are not added back).

As shown in *Figure 4*, neither the peptide- nor the pan-specific model benefitted from reusing the redundant data. In fact, the performance of the pan-specific model was significantly reduced in terms of unweighted AUC (p=0.0041) and weighted AUC 0.1 (p=0.0395). This is likely caused by the larger imbalance in observations per peptide introduced by the redundant data, as a large proportion of these observations came from the already abundant GILGFVFTL peptide.

## Removing potential outliers from training leads to better performance

During the testing of our models, we observed that several peptides consistently had a performance much lower compared to other peptides characterized with similar amounts of data. One thing shared by these peptides is that 10 X sequencing made up the vast majority of the experimental source of the recorded TCRs, as shown in *Table 1*. For most of the peptides with poor performance (KLG, AVF, IVT, RLR, RLP, SLF), only 10 X sequencing data was available. On the other hand, not all 10 X data are bad, as illustrated by RAKFKQLL which is a high performing peptide only covered by 10 X data (see for instance *Supplementary file 1*). Further, when comparing the predicted score distributions between positive and negative TCRs, we observe examples of outliers with low scoring positive TCRs and high scoring negative TCRs across all peptides (see *Figure 5—figure supplement 1*). These observations strongly suggest that the data contain a certain degree of wrongly labeled entries, and that these could be a source to limit the performance of the models. Inspired by the plot in *Figure 5—figure supplement 1*, outliers were identified by scoring TCRs using the NetTCR-2.2 peptide-specific model, and positive and negative TCR outliers assessed based percentile scores estimated from the contrary TCR pool (for details refer to Materials and methods). Using this approach, TCRs were removed from the training data based on percentile thresholds of 50%, 60%, 70%, 80%, 85%, 90%, and 95% respectively. That is, for a threshold of 70%, a positive TCR was identified as an outlier if it had a predicted score below the lower 70% percentile score range of the negative TCRs for all models predicting on the validation data (four models per partition). Next, pan-specific models were trained using the "limited" data for training and validation, while evaluating the models based on the full dataset.

An overall increase in performance for the models trained on the limited datasets was observed up until the 70th percentile datasets, after which the performance gain stagnated (see *Figure 5—figure supplement 2*). Since the difference in performance between the 80th and 70th percentile model was statistically insignificant for any of the bootstrap metrics (p>0.08 in all of weight and unweight performance metrics), the 70th percentile dataset for removing outliers from training was selected, since this filtering removed the least amount of data. As seen in *Figure 5—figure supplement 3*, more observations were, as expected, removed for the peptides with poor performance, indicating a higher presence of outliers for these peptides. The average performance of the model trained on the 70th percentile dataset was significantly higher than the model trained on the full dataset (p=0.0001, p<0.0001, p=0.0054 and p<0.0001 for AUC, weighted AUC, AUC 0.1 and weighted AUC 0.1, respectively). As shown in *Figure 5*, a higher performance was also consistently observed for the peptides which originated from 10 X sequencing, apart from the RLP peptide, which obtained a slightly lower AUC (−0.0066). While most peptides benefitted from the removal of potential outliers, some peptides did receive a substantially lower performance. It should however be noted that the performance evaluation was conducted on the full dataset, meaning that if a peptide has many actual outliers, the performance may be underestimated, since these outliers are included in the evaluation.

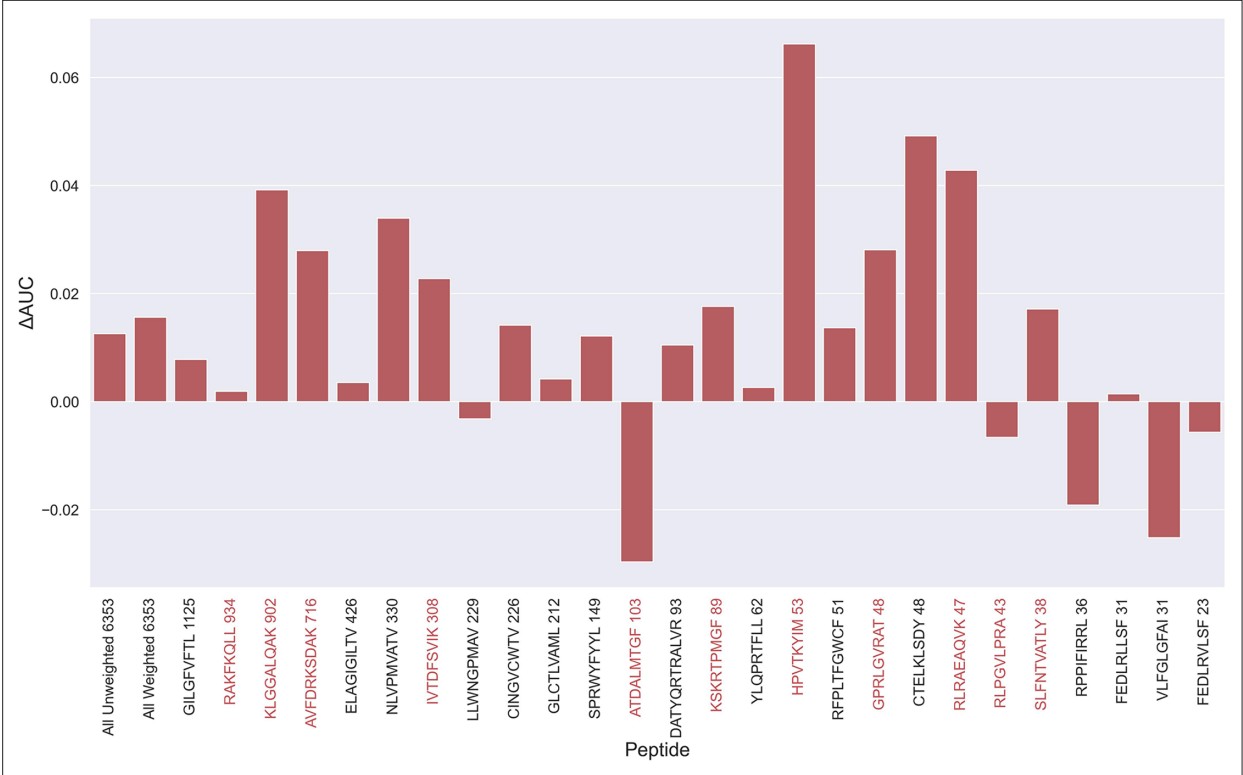

**Figure 5.** Difference in AUC between pan-specific CNN trained on the limited dataset (70th percentile) and full dataset. Peptides with TCRs originating solely from 10 x sequencing are highlighted in red. The performance was in both cases evaluated per peptide on the full dataset. A positive ΔAUC indicates that the model trained on the limited dataset performs better than the model trained on the full dataset. The performance differences are based on the performance of model 10 and model 4 in ***Supplementary file 1***, with model 4 being the baseline.

The online version of this article includes the following figure supplement(s) for figure 5:

**Figure supplement 1.** Prediction values on the full test data for each peptide when predicted using the NetTCR 2.2 - Peptide model.

**Figure supplement 2.** Mean AUC of the pan-specific NetTCR 2.2 models when trained on datasets with potential outliers removed.

**Figure supplement 3.** Percentage of observations discarded for the 70th percentile limited dataset, as a result of the removal of potential outliers.

## Improving the peptide-specific models

### Pre-training

As described earlier, the pan-specific model was generally observed to excel in terms of AUC, whereas the peptide-specific model was better in terms of AUC 0.1.

To benefit from the strengths of both of these models, a new model architecture was investigated. In brief, this architecture consists of two blocks of CNNs; one which is used for training on a pan-specific dataset to learn a general representation of binding, while the other block is used to train on a peptide-specific dataset to better learn the pattern of binding specific to a certain peptide (for details refer to Materials and methods). The pan-specific CNN block was trained first, with frozen initial weights and biases in the peptide-specific CNN block. After pre-training the pan-specific CNN block, these pan-specific CNN layers were frozen, whereas the layers for the peptide-specific CNN were allowed to update during the peptide-specific training.

As shown in ***Figure 6***, this pre-trained model outperformed both the pan- and peptide-specific models. This improvement was found to be highly significant (p<0.0001) across all metrics, when compared to the bootstrap of the pan-specific model, which was also the case when comparing to the peptide-specific model (p<0.0001, p<0.0001, p=0.0008 and p=0.0021 for AUC, weighted AUC, AUC 0.1 and weighted AUC 0.1, respectively). Furthermore, this pre-trained model had higher performance across all metrics than a simple ensemble of the pan-specific and peptide-specific models (data not shown).

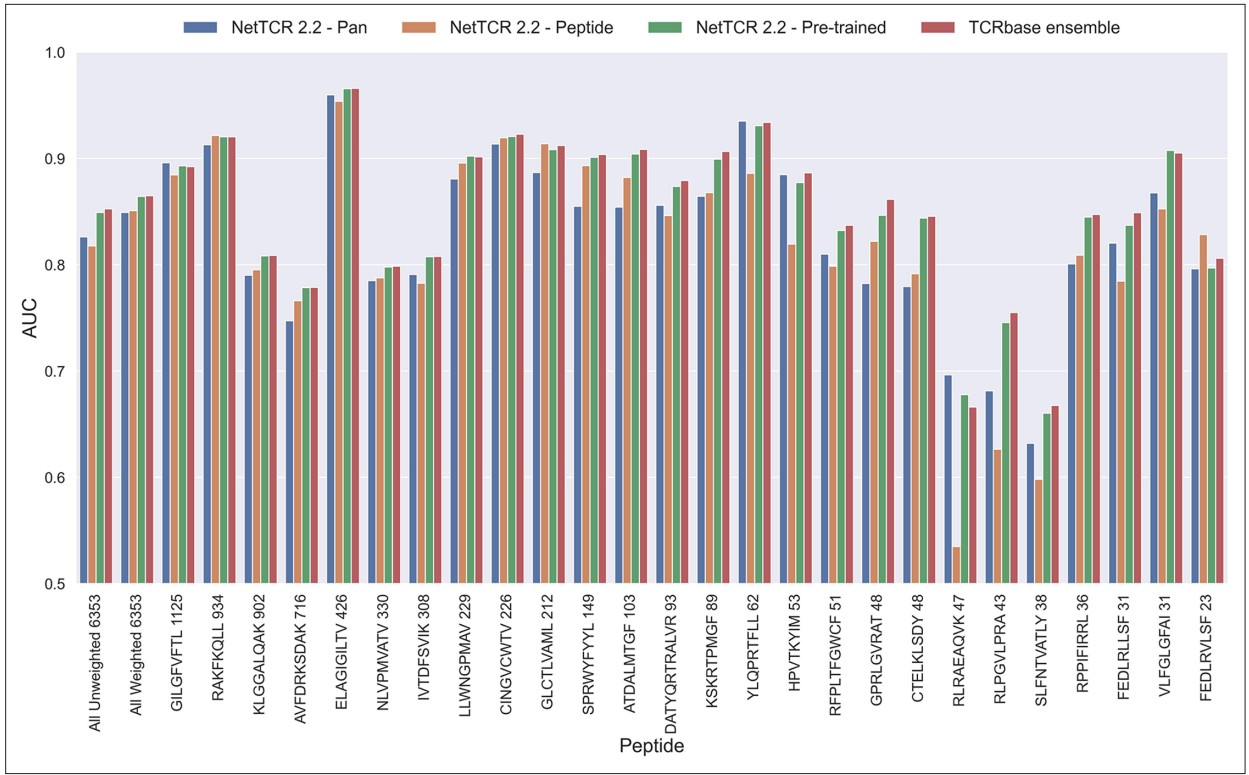

**Figure 6.** Per peptide performance of the updated peptide-specific, pan-specific, and pre-trained CNN in terms of AUC, when trained on the limited training dataset and evaluated on the full dataset. The peptides are sorted based on the number of positive observations from most abundant to least abundant, with the number of positive observations listed next to the peptide sequence. The unweighted (direct) mean of AUC across all peptides is shown furthest to the left, while the weighted mean is shown second furthest to the left. The weighted mean is weighted by the number of positive observations per peptide and puts more emphasis on the peptides with the most observations. The models included in this figure corresponds to model 10 (NetTCR 2.2 - Pan), model 15 (NetTCR 2.2 - Peptide), model 16 (NetTCR 2.2 - Pre-trained) and model 17 (TCRbase ensemble) in *Supplementary file 1*.

The online version of this article includes the following figure supplement(s) for figure 6:

**Figure supplement 1.** Per peptide performance of the updated peptide-specific, pan-specific, and pre-trained CNN in terms of AUC 0.1, when trained on the limited training dataset and evaluated on the full dataset.

## TCRbase ensemble

Earlier work has demonstrated a high performance of simple similarity-based models for prediction of TCR-specificity (*Meysman et al., 2023*). We therefore wanted to investigate if the predictive power could be further improved by integrating the sequence-similarity based predictions of TCRbase (*Montemurro et al., 2022*) into our modeling framework. In short, TCRbase makes predictions by calculating a similarity between a given TCR and the positive TCRs for a given peptide in terms of a sum over the paired similarities over the 6 CDR loops (*Montemurro et al., 2022*). TCRbase was integrated in terms of a simple scaling factor so that the pre-trained CNN model predictions were multiplied by the TCRbase predictions lifted to a power of $\alpha>0$. The optimal value of α was here estimated based on the validation partitions, and the test partitions were removed from the positive database given to TCRbase, to avoid overfitting and performance overestimation.

As shown in *Figure 7a*, the use of TCRbase predictions as a scaling factor resulted in a consistent increase in performance across both unweighted mean AUC and AUC 0.1. Although the mean AUC was only affected slightly by this scaling (maximum increase of 0.00212 at $\alpha=14$), a greater increase in performance was observed in terms of AUC 0.1 (maximum increase of 0.00723 at $\alpha=8$). Overall, the integration of TCRbase led to a significant improvement in performance for all metrics (p<0.0001). It should however be noted that while the use of the TCRbase scaling generally improved performance, the optimal α factor varied between each validation partition and cross-validation model (see *Figure 7b*). Nevertheless, the median of the optimal α across the cross-validation models was

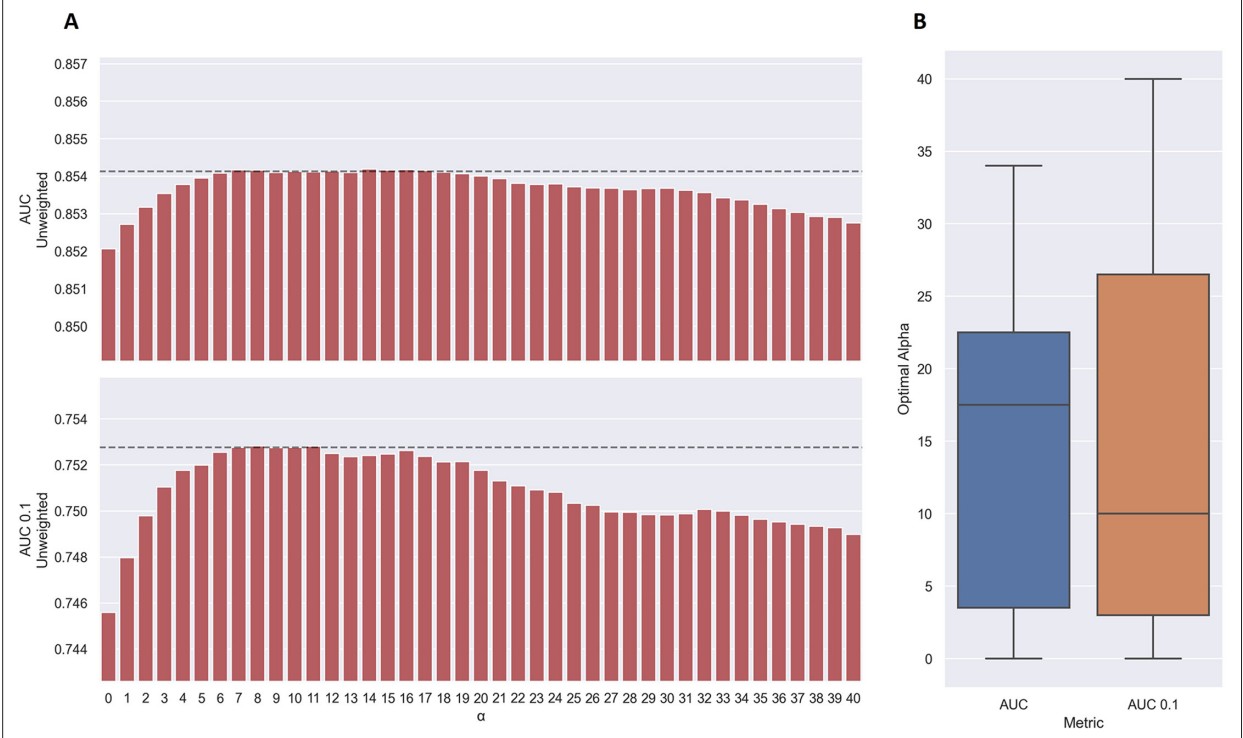

**Figure 7.** Performance of TCRbase ensemble as a function of α along with boxplot of optimal alpha in terms of AUC and AUC 0.1 for the validation partitions. (**A**) The predictions of the pre-trained model ensemble (trained on the limited dataset) on the test partitions (full data) were scaled by the kernel similarity to known binders, as given by TCRbase with a weight of (1,1,3,1,1,3), to a power of α. The performance is given as the unweighted mean performance across all 26 peptides, in terms of AUC and AUC 0.1. The dashed line shows the performance when α is set to 10, which strikes a good balance between AUC and AUC 0.1. An α of zero corresponds to the model ensemble without the TCRbase scaling. (**B**) Boxplot of the optimal alpha scaling factor per cross-validation model, when evaluated in terms of AUC and AUC 0.1, respectively, on the validation partitions. The models used for calculating the performance of the ensembles in this figure are model 16 (NetTCR 2.2 - Pre-trained) and model 21 (TCRbase) in **Supplementary file 1**.

The online version of this article includes the following figure supplement(s) for figure 7:

**Figure supplement 1.** Difference in true positive rate (TPR) between TCRbase ensemble (pre-trained +TCRbase models) and pre-trained models as a function of false positive rate (FPR).

10 in the case of AUC 0.1, which strengthened our confidence in using this α as the base scaling factor. Despite these variable observations, we for the sake of consistency stick to an α of 10 for the remaining analysis in this paper. The performance per peptide when using the $\alpha=10$ scaling is shown in **Figure 6** (AUC), as well as **Figure 6—figure supplement 1**.

We also observed that the optimal value for α varied between peptides, with a slight positive correlation to the performance of TCRbase for the given peptide (see **Table 3**), suggesting the peptides with high TCRbase performance benefit more from the α rescaling.

**Table 3.** Pearson Correlation Coefficients (PCC) between the optimal α scaling factor and performance per peptide in terms of AUC and AUC 0.1 of the pre-trained CNN model and TCRbase model, respectively, for the validation partitions.

Each partition was considered as a separate sample. p-Values for the null hypothesis that the performance and optimal α are uncorrelated are also shown.

| Metric | PCC to optimal alpha | p-Value |
|---|---|---|
| CNN AUC | –0.1101 | 0.2123 |
| TCRbase AUC | 0.3056 | 0.0004 |
| CNN AUC 0.1 | –0.0809 | 0.3602 |
| TCRbase AUC 0.1 | 0.2068 | 0.0183 |

To investigate further how the integration of TCRbase predictions benefitted the performance, we in *Figure 7—figure supplement 1* plotted the difference in true positive rates at different false positives rates between the TCRbase ensemble and the pre-trained CNN alone. This figure demonstrates that the benefit from TCRbase mainly consist of increasing the discrimination between binders and non-binders at thresholds corresponding to low FPRs ($0 \leq$ FPR $<= 0.15$), whereas the predictions may become slightly worse than without scaling when the threshold for binders are set to that of an FPR higher than 0.3. This result thus suggests that scaling the predictions of neural network models based on similarity to known binders is mostly beneficial when a high specificity is desired.

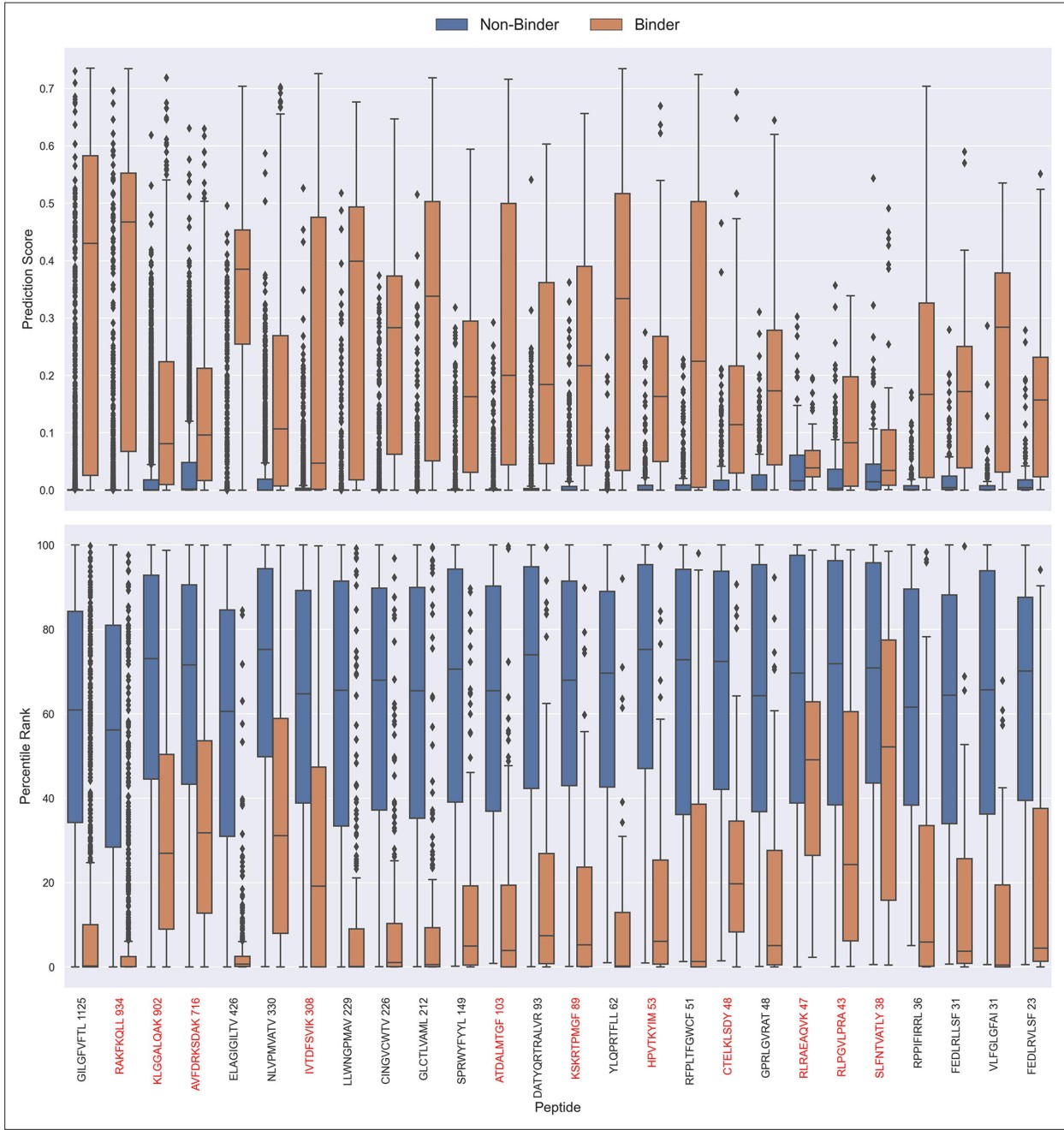

**Figure 8.** Boxplot of direct prediction scores and percentile ranks per peptide of the full test dataset for the TCRbase ensemble. Peptides with 100% of positive observations coming from 10 X sequencing are highlighted in red. The model used in this figure is model 17 (TCRbase ensemble) in *Supplementary file 1*.

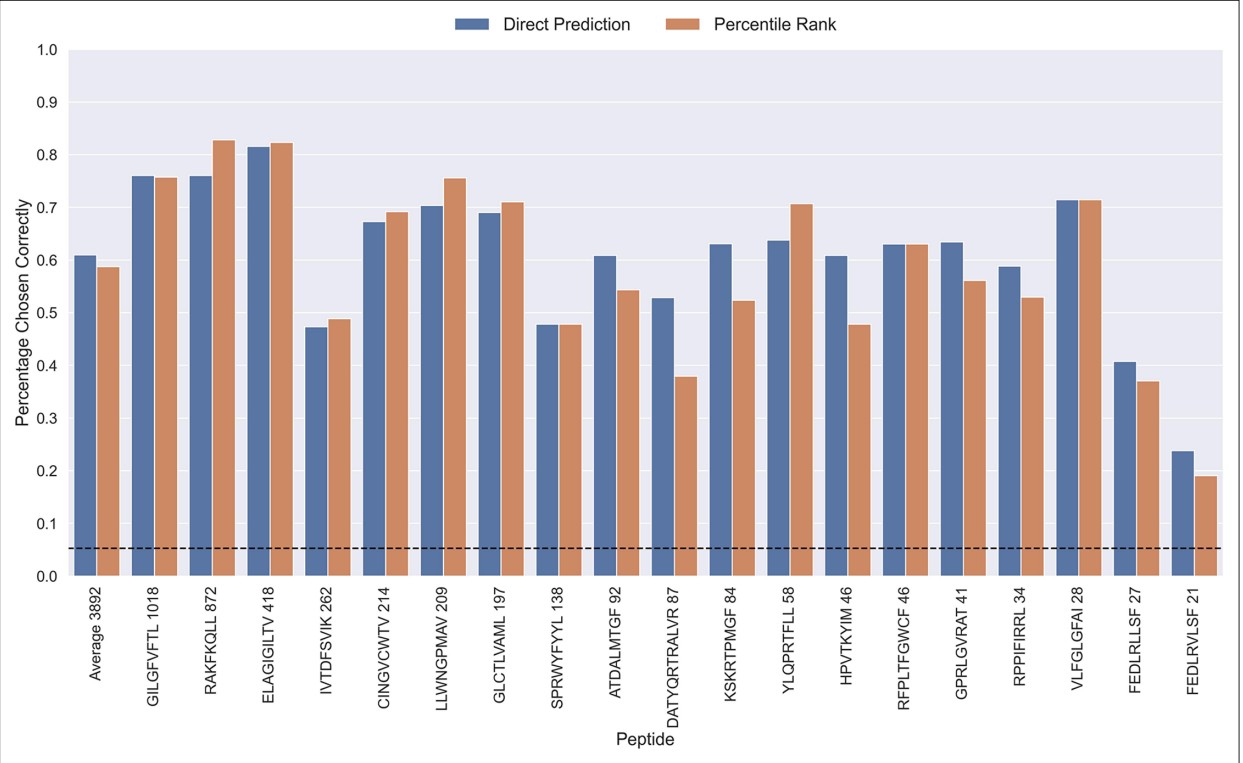

**Figure 9.** Percentage of correctly chosen true peptide-TCR pairs for each peptide in the limited dataset. This was evaluated using the direct prediction score (blue) and the percentile rank (orange) of the TCRbase ensemble. KLGGALQAK, AVFDRKSDAK, NLVPMVATV, CTELKLSDY, RLRAEAQVK, RLPGVLPRA, and SLFNTVATLY were excluded from this analysis due to low predictive performance for these peptides (AUC 0.1<0.65). The numbers next to the peptides indicate the number of positive TCRs in the filtered dataset, and the dashed line indicates the expected value for a random prediction. The predictions are based on model 17 (TCRbase ensemble) in *Supplementary file 1*.

The online version of this article includes the following figure supplement(s) for figure 9:

**Figure supplement 1.** Boxplot of average rank per peptide for the final updated models.

## Percentile rank rescaling

The prediction scores of the final CNN + TCRbase model ensemble fall between 0 and 1 but display substantial score distribution variations between peptides (see *Figure 8*), which makes it hard to directly compare prediction scores between peptides. A common approach to resolve this is to apply percentile rank scores (*Montemurro et al., 2022*). Here, we used the CNN + TCRbase model to predict scores for a set of 15,957 negative TCRs for each peptide, which was obtained from the dataset for the IMMREP 2022 workshop (*Meysman et al., 2023*), and used these scores to calculate a percentile rank for each observation in our test data. Here, the percentile rank is defined as the proportion (in percentage) of negative controls, which scored higher than the given observation. As shown in *Figure 8*, the percentile ranks for binders between peptides are more similar when compared to the direct prediction scores.

## Peptide specificity test

The performance evaluations performed so far have focused on the ability to predict whether or not a TCR can bind to a given peptide. Another important aspect is the ability to predict the correct peptide target of a given TCR. To investigate the performance in this context, each positive TCR was scored against all peptides, and a performance metric was estimated in terms of how often the correct TCR-peptide pair was given the highest score (or lowest percentile rank). The result of this analysis is shown in *Figure 9*, which was conducted on the limited dataset, while excluding observations for low performing peptides with an AUC <0.8 and AUC 0.1<0.65 for the TCRbase ensemble (see *Figure 6* and *Figure 6—figure supplement 1*). This dataset thus consists of 21 peptides, and a random predictor is expected to obtain a performance of 1/19~0.05. The results show that the model

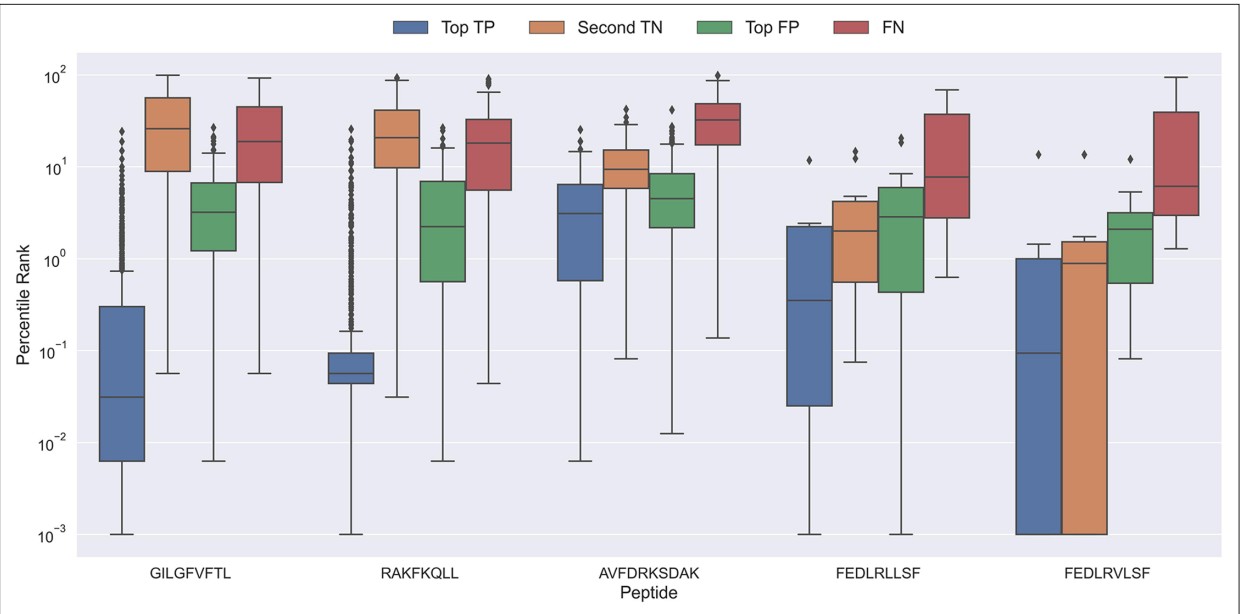

**Figure 10.** Boxplot of percentile ranks per peptide in the rank test, with KLGGALQAK, NLVPMVATV, CTELKLSDY, RLRAEAQVK, RLPGVLPRA, and SLFNTVATLY excluded. AVFDRKSDAK was included as an example of a peptide with a poor rank in the rank test. Top TP: Percentile rank of the correctly chosen pairs. Second TN: Percentile rank of the second-best pair, when the correct pair was chosen. Top FP: Percentile rank of the best scoring pair when the incorrect pair was chosen. FN: Percentile rank of the correct pair, when the incorrect pair was chosen. The predictions are based on model 17 in **Supplementary file 1**.

clearly outperforms this random baseline for all peptides. Also, a higher performance is observed for the three most abundant peptides in this analysis (GILGFVFTL, RAKFKQLL and ELAGIGILTV). Furthermore, it is seen that there is a slight tendency for a lower percentage of correctly chosen peptide-TCR pairs, as the number of positive TCRs for the training becomes lower. Interestingly, the percentage of correctly chosen pairs correlates very strongly with the AUC and the AUC 0.1 of the peptides. In the case of ranks when using direct prediction, the PCC of the percentage of correct predictions to AUC and AUC 0.1 were 0.740 and 0.830, respectively (sample size of 19 peptides). This high correlation was also observed for percentile ranks, with a PCC of 0.706 and 0.873 to AUC and AUC 0.1, respectively.

Furthermore, a tendency of lower average ranks for the pre-trained and TCRbase ensemble models compared to the other models was observed (see **Figure 9—figure supplement 1**). However, while the application of percentile rank widened the range of average rank per peptide, it generally resulted in a decrease of the median rank for most peptides.

To better understand why the models sometimes failed to predict the correct peptide-TCR pair, we looked at the distribution in percentile rank of the top scoring pairs. As **Figure 10** shows, the binding TCRs for the peptides with a high proportion of correctly chosen pairs (GILGFVFTL and RAKFKQLL) is characterized by having a low percentile rank of around 0.1 (see 'Top TP').

Peptides which had poor predictive performance (mainly those excluded in **Figure 9**) generally had a poor specificity with less than 20% peptide-TCR pairs chosen correctly. These peptides are characterized by having a much higher percentile rank for the true peptide:TCR pair, as exemplified by AVFDRKSDAK in **Figure 10**, which again indicates the presence of potential outliers for these peptides.

Interestingly, the percentile ranks for the TCR pairs of two of the peptides FEDLRLLSF and FEDL-RVLSF are characterized by having a very low percentile rank for the second highest scoring pair. This appears to be at least partially due to a high shared similarity between these two peptides, causing the model to mislabel the top scoring peptide. For example, for FEDLRLLSF, the best scoring peptide was FEDLRVLSF 22.2% and 25.9% of the time for predictions and percentile ranks, respectively. For FEDLRVLSF, the best scoring peptide was FEDLRLLSF 23.8% and 38.1% of the time for predictions and percentile ranks, respectively.

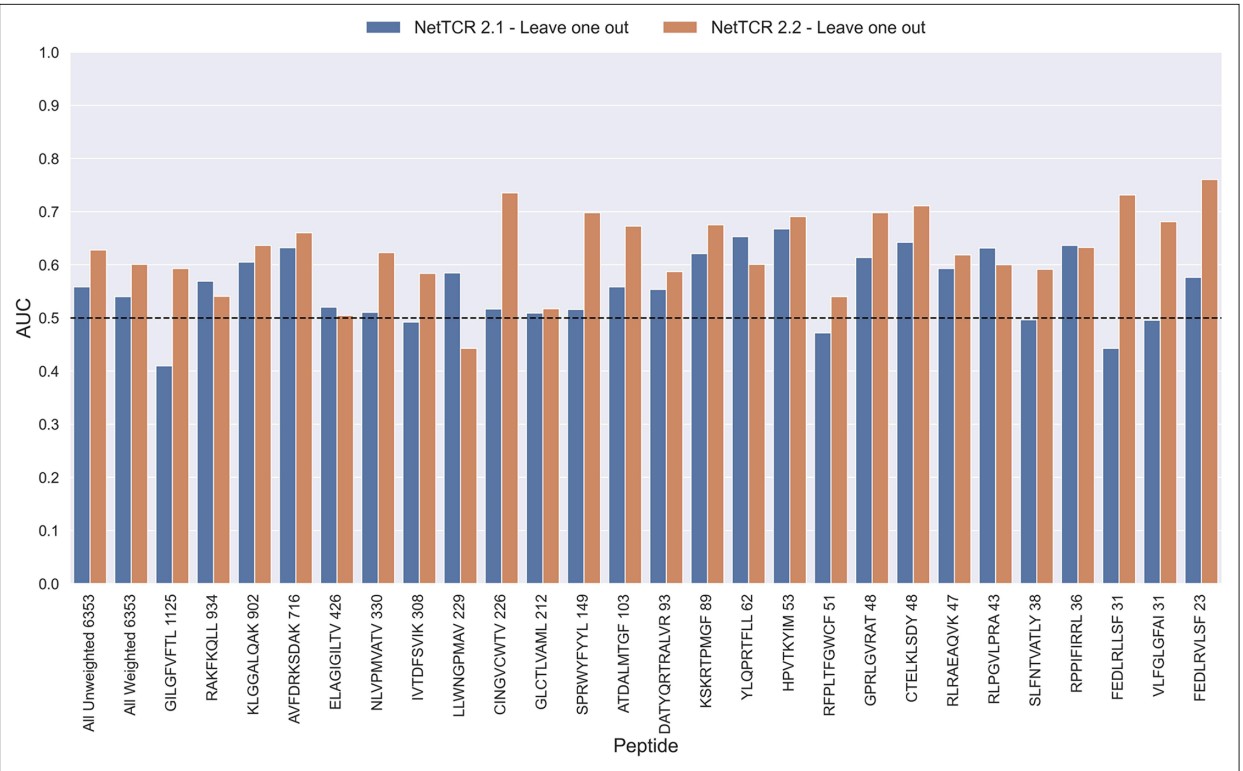

**Figure 11.** Per peptide performance of the old (NetTCR 2.1) and updated (NetTCR 2.2) pan-specific CNN models trained in a leave-one-out setup. The performance was evaluated in terms of AUC on the full dataset. The performance shown in this figure is based on model 63 (NetTCR 2.1 - Leave one out) and model 19 (NetTCR 2.2 - Leave one out) in *Supplementary file 1*.

The online version of this article includes the following figure supplement(s) for figure 11:

**Figure supplement 1.** Per peptide performance of the old (NetTCR 2.1) and updated (NetTCR 2.2) pan-specific CNN models trained in a leave-one-out setup.

Generally, it should also be noted that in cases where the correct peptide-TCR is not given the lowest rank, the correct peptide-TCR pair is given a very high percentile rank, most often greater than 20 (refer to FN label in *Figure 10*). The same observation holds for the top scoring peptides in these cases (top FP in *Figure 10*). This once again indicates that there might be some potential wrongly labeled outliers in the positive data, even when the data is filtered with the use of the model predictions.

## Performance when data is scarce or absent

Having demonstrated a robust and high performance of the CNN-pan-specific model in the context of TCR specificity towards known peptides, i.e. peptides included in the training data, we next turned to the uttermost challenging question namely prediction of TCR specificity towards novel peptides.

To investigate this, we trained models in a pan-specific leave-one-out setup, where for a given peptide, both positives and negatives generated from that peptide were removed from the training data, thus preventing data leakage. This was done both for the NetTCR 2.1 and the updated NetTCR 2.2 architecture. For this experiment, the limited training dataset with outliers removed was used. This resulted in 26 different models, each of which was evaluated on the peptide dataset for the left-out peptide. As shown in *Figure 11*, a performance in terms of AUC slightly better than random was observed for most of the peptides. Furthermore, a noticeable improvement in performance was seen for the updated NetTCR 2.2 model. However, the performance was almost completely random when evaluated in terms of AUC 0.1, as can be seen in *Figure 11—figure supplement 1*. The only peptides with non-random AUC 0.1 performance were FEDLRLLSF and FEDLRVLSF, and this was only the case for the NetTCR-2.2 model architecture. These peptides differ by only a single amino acid, and the

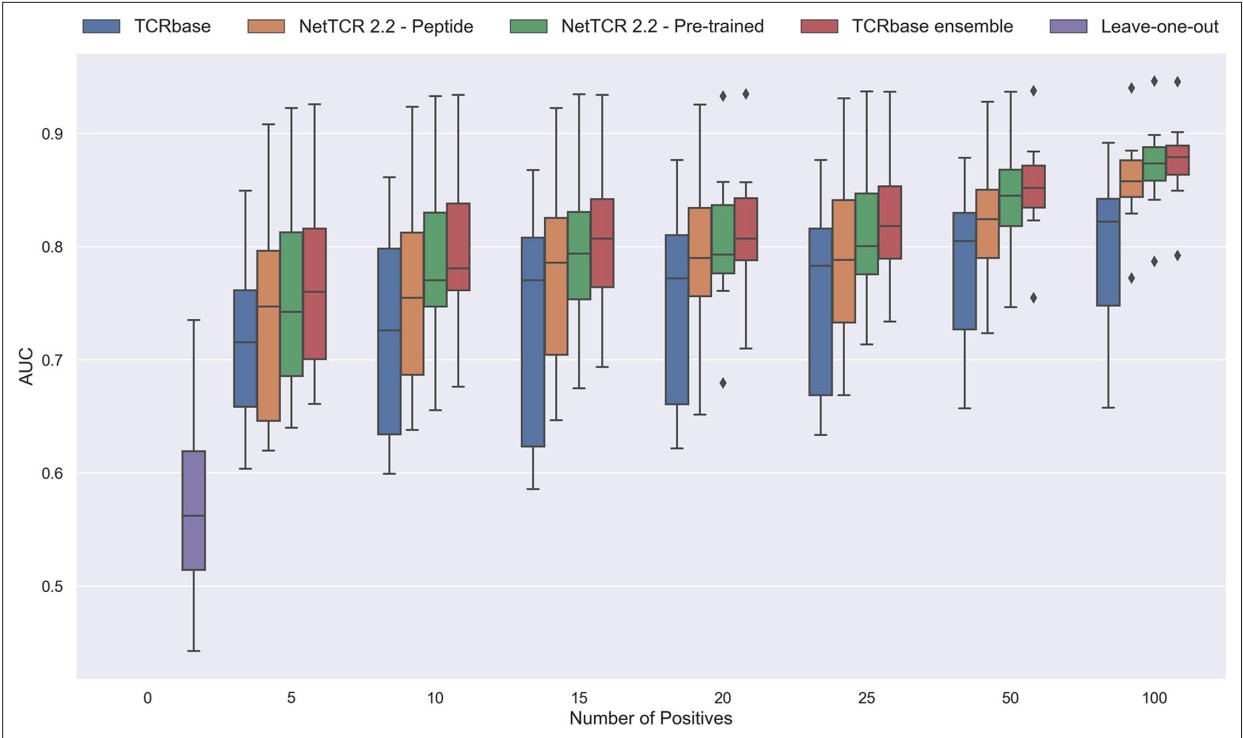

**Figure 12.** Performance in terms of AUC of various models trained on increasing amounts of data. These models were trained on the following peptides: GILGFVFTL, RAKFKQLL, ELAGIGILTV, IVTDFSVIK, LLWNGPMAV, CINGVCWTV, GLCTLVAML, and SPRWYFYYL. The pre-trained models were based on the leave-one-out model, and afterwards fine-tuned and re-trained on the smaller training datasets. The performance shown is based on the predictions for model 24–51 in *Supplementary file 1*.

The online version of this article includes the following figure supplement(s) for figure 12:

**Figure supplement 1.** Performance in terms of AUC 0.1 of various models trained on increasing amounts of data.

result thus indicates that the updated model in this case is able to transfer the knowledge gained from training on another similar peptide, which was not the case with the old architecture.

We next extended the analysis to a leave-most-out setting to investigate how little data is required in order to train models with non-random performance. Here, a number of training datasets were generated by subsampling the limited dataset in order to achieve 5, 10, 15, 20, 25, 50, and 100 positive observations, respectively, per peptide. Swapped negatives were also subsample in this way, keeping a ratio of 1:5 between binders and non-binders. This was only done for the peptides GILG-FVFTL, RAKFKQLL, ELAGIGILTV, IVTDFSVIK, LLWNGPMAV, CINGVCWTV, GLCTLVAML and SPRWY-FYYL, since they all had substantial performance (AUC 0.1 ≥ 0.65) for the full model and more than 100 positive observations to begin with.

In the case of the pre-trained model, the leave-one-out model was used as the startpoint. Rather than having to re-train the full pan-specific CNN block (which may be impractical, if a user wants to re-train the model on a new peptide), we decided to instead fine-tune this CNN block by adding the subsampled data to the leave-one-out training data, while setting the sample weight to 1 for the new peptide observations, and 0.1 for the remaining observations. The pan-specific CNN block was then trained for 30 epochs in this way (for details refer to Materials and methods).

As shown in *Figure 12* and *Figure 12—figure supplement 1* all models demonstrated a non-random performance with as low as 5 positive observations. As expected, a general increase in performance was observed as more and more data was available for training. This was especially the case for the TCRbase ensemble model, which strongly outperformed all other models with an AUC close to 0.8, when the number of training points surpassed 15.

Noticeably, the performance of the baseline TCRbase model did not improve nearly as much as the CNN-based models when the amount of training data was increased, suggesting that the CNN

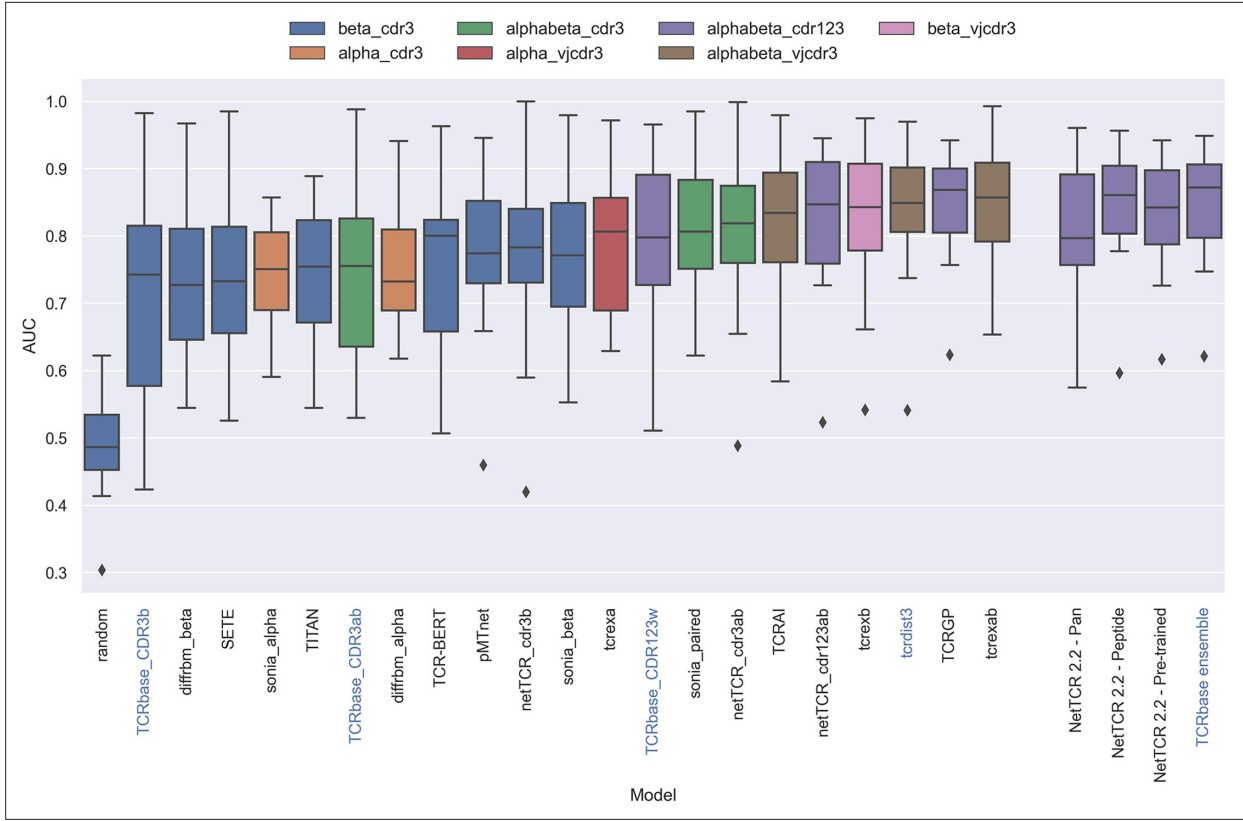

**Figure 13.** Boxplot of reported unweighted AUC per peptide for the models in the IMMREP benchmark, as well as the updated NetTCR 2.2 models. Except for the updated NetTCR 2.2 models (NetTCR 2.2 - Pan, NetTCR 2.2 - Peptide, NetTCR 2.2 - Pre-trained and TCRbase ensemble) the performance of all models is equal to the reported performance in the IMMREP benchmark. The color of the bars indicates the type of input used by the model. Machine-learning models are labeled with black text, whereas distance-based models are labeled with blue text. Note that the TCRbase ensemble is a mixture between a machine-learning and distance-based model. The performance of the NetTCR 2.2 models is based on model 53 (NetTCR 2.2 - Pan), model 54 (NetTCR 2.2 - Peptide), model 55 (NetTCR 2.2 - Pre-trained), and model 56 (TCRbase ensemble) in *Supplementary file 2*. The performance of the remaining models are based on the values listed in the IMMREP 2022 GitHub repository at https://github.com/viragbioinfo/IMMREP_2022_TCRSpecificity/blob/main/evaluation/microaucs.csv.

The online version of this article includes the following figure supplement(s) for figure 13:

**Figure supplement 1.** Boxplot of average rank per peptide per model in the IMMREP test data, as reported in the IMMREP benchmark.

models are able to benefit much more from the increased amount of information present in larger datasets.

## External evaluation
### IMMREP 2022 benchmark
Having defined a novel and improved architecture and framework for training models for prediction of TCR specificity, we next turned to an independent data set to confirm its robustness. Here, we applied the datasets from the IMMREP 2022 workshop (*Meysman et al., 2023*), keeping all model hyperparameters unchanged compared to the different models described above. As shown in *Figure 13*, the updated peptide-specific models, NetTCR-2.2 - Peptide, significantly outperformed NetTCR 2.1 (p=0.0367, p=0.0263, p=0.0087 and p=0.0034 for AUC, weighted AUC, AUC 0.1 and weighted AUC 0.1, respectively). With an unweighted average AUC of 0.8476, this model performed on par with the best performing model in terms of AUC at the IMMREP workshop, TCRex αβ (*Gielis et al., 2018*), with an average unweighted AUC of 0.8473. However, to our surprise, and contrary to our findings on the original dataset of this paper, the NetTCR-2.2 - Pre-trained model underperformed compared to the peptide-specific model, even though part of this performance loss was recovered when introducing the TCRbase scaling on the pre-trained model. Furthermore, the NetTCR-2.2 - Pan model was also found to perform much worse than expected.

**Table 4.** Degree of redundancy between the IMMREP test and training data, when using a 95% kernel similarity threshold for redundancy within each peptide.

The redundancy reduction was performed on both positive and negative observations. The counts and percentages, however, only refers to the positive observations.

| Peptide | Pre reduction count | Post reduction count | Percent redundant |
|---|---|---|---|
| All | 619 | 467 | 24.56% |
| GILGFVFTL | 136 | 58 | 57.35% |
| NLVPMVATV | 69 | 54 | 21.74% |
| YLQPRTFLL | 67 | 53 | 20.90% |
| TTDPSFLGRY | 49 | 47 | 4.08% |
| LLWNGPMAV | 47 | 44 | 6.38% |
| CINGVCWTV | 46 | 46 | 0.00% |
| GLCTLVAML | 37 | 23 | 37.84% |
| ATDALMTGF | 26 | 22 | 15.38% |
| LTDEMIAQY | 25 | 23 | 8.00% |
| SPRWYFYYL | 24 | 24 | 0.00% |
| KSKRTPMGF | 22 | 13 | 40.91% |
| NQKLIANQF | 15 | 15 | 0.00% |
| TPRVTGGGAM | 12 | 12 | 0.00% |
| HPVTKYIM | 12 | 10 | 16.67% |
| NYNYLYRLF | 12 | 9 | 25.00% |
| GPRLGVRAT | 11 | 11 | 0.00% |
| RAQAPPPSW | 9 | 3 | 66.67% |

We further evaluated the peptide specificity of the models by calculating the average rank of each peptide in the benchmark specificity test dataset, and compared the ranks to those of the other methods included in the IMMREP benchmark. As is shown in *Figure 13—figure supplement 1*, also here the average ranks of the updated models were found to be comparable to the best performing models in the IMMREP benchmark.

To understand the source of the relatively poor performance of the pan-specific models in this benchmark, we further investigated the IMMREP datasets. Even though the construction of IMMRep datasets was made to ensure that no positive TCR was shared between the training and test data sets, inspection of the data revealed that swapped negatives were present in the training data, which originated from positive peptides in the test data. When a pan-specific model is trained on such data, this results in certain TCRs being 'seen' only as non-binders only during the model training. Given this, the model will likely assign such TCRs as negative when asked to predict the test data. This problem is limited to pan-specific models hence explaining the reduced performance compared to the peptide-specific model.

Further, as shown in *Table 4*, the degree of redundancy between training and test data was relatively high for many of the peptides. This redundancy between the IMMREP test and training data may result in test performance overestimation since the models observe similar TCR-peptide combinations during training.

When comparing the per peptide AUC of all models to the per peptide redundancy between training- and test data, we observed a Pearson correlation of 0.428 (sample size of 370), which was a much stronger correlation than observed between the number of training observations and AUC (0.062).

To address these issues, we applied the redundancy reduction and swapped negative data generation (generating the swapped within each data partition) from our own data pipeline on the training

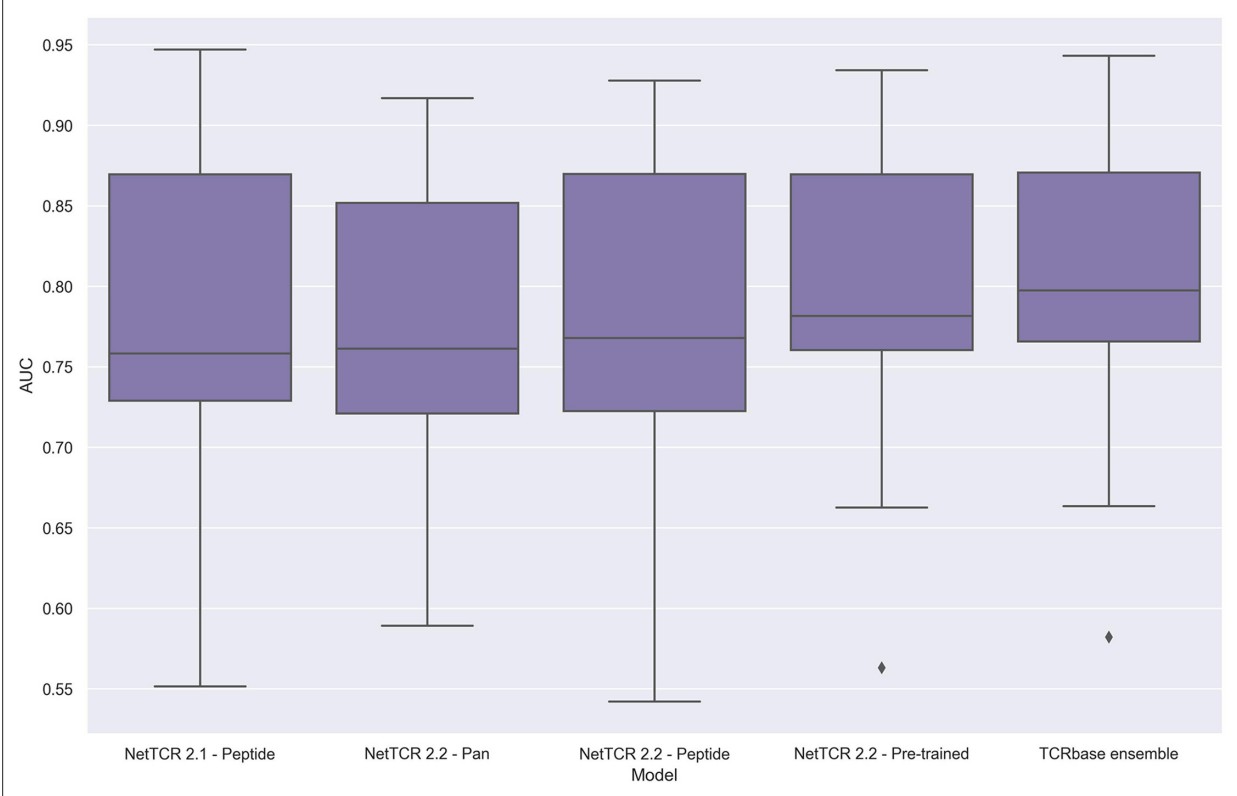

**Figure 14.** Boxplot of unweighted AUC per peptide for the NetTCR 2.1 and 2.2 models, when trained and evaluated on the redundancy reduced dataset. The evaluation was performed using a nested cross-validation setup. The performance is based on model 58 (NetTCR 2.1 - Peptide), model 59 (NetTCR 2.2 - Pan), model 60 (NetTCR 2.2 - Peptide), model 61 (NetTCR 2.2 - Pre-trained), and model 62 (TCRbase ensemble) in **Supplementary file 2**.

data, while ensuring a positive to swapped negative ratio of 1:3 and positive to negative control ratio of 1:2, as was the case for the original IMMREP dataset. The performance of the different models was next assessed via nested cross-validation on this dataset, rather than the original left-out test data. As shown in **Figure 14**, this data setup once again resulted in the pre-trained models outperforming the peptide-specific models, and that the use of TCRbase scaling together with the pre-trained model resulted in the overall best performance, in line with our earlier findings. Moreover, the overall performance of the models was found to drop, especially for the peptides with a high degree of redundant data, confirming a degree of performance overestimation in the original benchmark (see **Supplementary file 2** for individual peptide performance).

## Discussion

Here, we have presented an improved NetTCR framework for prediction of TCR specificity including updates to the training data, modeling architecture and training setup, with the goal of increasing the overall performance and generalization power of the model. First and foremost, the update includes a substantial expansion of the training data to 26 peptides, up from the six peptides available for predictions in NetTCR 2.1. The model updates included dropout and peptide-specific sample weights to deal with data imbalance, forcing the model to focus more evenly on all peptides, and resulted in vastly improved performance in the pan-specific setup. This performance gain was particularly pronounced for peptides with few observed binding TCRs. The updated architecture in the form of more hidden units in the dense layer, the change from sigmoid to ReLu activation for the max-pooling, and the introduction of dropout further improved the NetTCR model. A variation of the updated architecture was also investigated, which combined the properties of the pan-specific and peptide-specific models, by having two separate CNN blocks where one block was pre-trained separately in pan-specific setup, followed by training the second block in a peptide-specific setup. This pre-training setup resulted in an additional increase in performance, mainly for the least abundant peptides.

## How to best use available data for training

The scarceness of paired TCR data means that it often could be tempting to include all available data to the fullest, and include all redundant data for training. However, as we show here, the addition of redundant data in the training does not lead to improved performance. In fact, we found that the addition of redundant data may cause pan-specific models to underperform if the peptide imbalance of data is not accounted for, since the inclusion of redundant data often results in a further increased peptide imbalance.

The observation that the predictive performance for some peptides was much lower than expected given the amount of available training data, led us to believe that outliers in the form of false positives might be a potential issue. Furthermore, many of these peptides had in common that the main source of data was 10 X sequencing (*10x Genomics, 2020*), a platform known to have a high proportion of false annotations (*Zhang et al., 2021*; *Povlsen et al., 2023*). To deal with this issue, we implemented a machine learning driven approach for outlier detection using the predictions of the peptide-specific NetTCR models to identify observations which repeatedly received very poor predictions. The removal of these potential outliers from the training led to significantly improved test performance. It should also be noted that the data applied in the study included denoising for most of the 10 X data in the form of ITRAP (*Povlsen et al., 2023*), which together with ICON *Zhang et al., 2021* have earlier been shown to properly remove outliers (*Montemurro et al., 2023*). Nevertheless, our results suggest that some outliers had escaped these denoising steps, indicating that denoising methods should still be improved upon. While the use of our model predictions to remove outliers resulted in improved performance, we believe that this approach should only be considered a proof-of-concept, and that more elaborate ways to identify outliers merit further investigation.

## Integrating distance-based methods can improve performance of ML models

Inspired by the observation that sequence similarity distance-based models often achieve very high performance for the prediction of TCR specificity (*Meysman et al., 2023*), we investigated if integrating TCRbase predictions could improve the performance of our models. We integrated TCRbase by scaling the CNN prediction with the TCRbase prediction to a power of α, and found that the performance of this ensemble (Pre-trained +TCRbase) achieved a significantly improved performance in terms of AUC and AUC 0.1. Interestingly, further inspections revealed that the increased performance mainly resulted from improved discrimination of binders and non-binders when the binding-threshold was set to result in a low FPR. Given how we are often interested in keeping the FPR very low for TCR specificity predictions, the simple integration of TCRbase can thus vastly benefit many real-world use-cases for TCR specificity predictions. While we decided to use a general α of 10 for the TCRbase scaling, it is possible that performance could be improved further, if α is allowed to be flexible depending on the peptide. For example, one could imagine that a peptide with a very high TCRbase predictive performance could benefit from a higher α, compared to another peptide with a lower TCRbase performance. Furthermore, the amount of positive data also influences which α is optimal. It would therefore be interesting to further investigate this, as this could potentially lead to further improved performance. Finally, investigating the relation between TPR and FPR at different values of α could also benefit many actual use-cases, considering that the optimal alpha value could be determined based on the desired maximum FPR rate.

## Predictions for unseen peptide

It has repeatedly been shown that predicting TCR specificity for "unseen" peptides is extremely hard, especially for peptides that are very dissimilar to the peptides included in the training data (*Moris et al., 2021*; *Grazioli et al., 2022*). Investigating the performance of the pan-specific models in a leave-one-out setup revealed that the performance on unseen peptides overall remained very poor, also for the updated NetTCR-2.2 model. While the performance for NetTCR-2.2 in terms of AUC was generally better than random, the performance in terms of AUC 0.1 was very close to random, severely limiting its general potential use. Nevertheless, we observed that the performance of NetTCR 2.2 in the leave-one-out setup was improved when compared to NetTCR 2.1, especially for two peptides sharing a high mutual similarity. While the performance for these two peptides was still

low compared to that observed in the full training setup, this result affirms that given a broad enough peptide coverage, pan-specific models have the potential to predict binding also for unseen peptides.

## Improved performance when data is scarce

While high performance for unseen peptides so far remains very challenging, another important issue is to boost performance for peptides with relatively few observations. Performing a leave-most-data-out, a substantial increase in performance was observed compared to the leave-out-out experiment with as little as five training observations, and already with 15 observations, a satisfactory performance was observed. This is in great contrast to earlier work, where a number of ~150 was suggested to be required for modeling TCR specificity (*Montemurro et al., 2021*). These results thus suggest that the pre-trained models can beneficially be used as seeds for the development of peptide-specific models allowing for rapid fine-tuning to new data.

We also observed that the TCRbase ensemble based on the pre-trained model consistently outperformed any of the other models, both when data was very scarce, but also as the amount of training data was increased, highlighting the benefits of integrating distance-based methods for predictions. As a final note, we would also expect that the discrepancy between the performance of the peptide-specific- and pre-trained model will become larger as the number of peptides to train on increases in the future, as a pan-specific CNN block trained on a more diverse dataset should allow for better generalization.

## Performance on IMMREP 2022 benchmark

To compare the updated models with other models for TCR specificity predictions, we applied the modeling framework to the dataset from the IMMREP 2022 benchmark (*Meysman et al., 2023*). Here, we observed that the updated peptide-specific model performed on par with the best models in the benchmark. We however also observed that the pre-trained model performed worse than expected. Careful inspection of the data revealed that swapped negatives had been generated across the test and training data, meaning that some TCRs were only seen as negatives in the training, whereas they could be positive in the test data, albeit for a different peptide. This problem strongly affected the pre-trained model, which had a pan-specific component. Furthermore, since redundancy was only dealt with by removing duplicate TCRs, redundancy in both training and test data was observed, resulting in a certain degree of performance overestimation. This was for instance reflected in an unusually high performance for the peptides which had higher degrees of redundancies between training and test data.

To deal with these problems, we performed redundancy reduction on the training data identical to what was done for our novel extended data set, and made sure to only generate swapped negatives from TCRs within a given partition. We then trained and evaluated our models using the nested cross-validation approach on this redundancy reduced data. Here, we recovered the earlier conclusion that the pre-trained models outperformed the peptide-specific models, and that the integration of TCRbase led to the highest overall performance. These results thus strongly underline a problematic issue with data redundancy and the leakage of swapped negative TCR between training and test datasets present in the IMMREP benchmark. This is of high concern, since these properties, as shown here, are in particular detrimental for pan-specific models. Considering this, we encourage the creation of a new benchmark which takes these issues into account, while ideally also expanding on the number of peptides present for predictions.

## Conclusion

In this work, we have demonstrated how prediction of TCR specificity can be greatly improved by introducing minor but critical updates to the NetTCR training and modeling framework. While also improving on the peptide-specific models, these updates in particular boost the performance of pan-specific models. In addition, we show that pre-training models on pan-specific data, followed by training in a peptide-specific setup, leads to substantially improved performance, especially when the amount of data is low. Scaling the predictions from NetTCR with similarity to known binders is also shown to boost performance. Further, we have for the first time demonstrated how machine learning models can be designed and applied for rational data denoising in the context of TCR specificity data. The performance for 'unseen' peptides was found to be overall low. However, the results

demonstrated an encouraging tendency of high predictive power in cases of 'unseen' peptides with high similarity to the training data.

## Additional information

### Funding

| Funder | Grant reference number | Author |
|---|---|---|
| Inno4Vac | 101007799 | Morten Nielsen |
| National Institute of Allergy and Infectious Diseases | 75N93019C00001 | Morten Nielsen |

The funders had no role in study design, data collection and interpretation, or the decision to submit the work for publication.

### Author contributions

Mathias Fynbo Jensen, Conceptualization, Data curation, Software, Formal analysis, Investigation, Visualization, Methodology, Writing – original draft, Project administration, Writing – review and editing; Morten Nielsen, Conceptualization, Supervision, Funding acquisition, Validation, Investigation, Methodology, Writing – original draft, Project administration, Writing – review and editing

### Author ORCIDs

Mathias Fynbo Jensen ⬤ http://orcid.org/0009-0004-6664-448X
Morten Nielsen ⬤ http://orcid.org/0000-0001-7885-4311

Reviewer #1 (Public Review): https://doi.org/10.7554/eLife.93934.3.sa1
Reviewer #2 (Public Review): https://doi.org/10.7554/eLife.93934.3.sa2
Author Response https://doi.org/10.7554/eLife.93934.3.sa3

## Additional files

### Supplementary files

• Supplementary file 1. Overview of training data, model parameters, predictions and performance of the models trained and evaluated in this article, excluding the models trained and evaluated on the IMMREP 2022 dataset. The listed Model Number for each model can be used to find the source data for the figures in this article (see the figure legends).

• Supplementary file 2. Overview of training data, model parameters, predictions and performance of the models trained and evaluated on the IMMREP 2022 dataset. The listed Model Number for each model can be used to find the source data for the figures in this article (see the figure legends).

• MDAR checklist

### Data availability

The final peptide-specific, pan-specific and pre-trained models, along with the main datasets, are available on GitHub at https://github.com/mnielLab/NetTCR-2.2, copy archived at *mnielLab, 2024* and a web server for the pan- and pre-trained models is available at https://services.healthtech.dtu.dk/services/NetTCR-2.2/, where an easy-to-use interface is provided for predictions. In addition, an overview of the performance of the different models, along with their predictions and training data, is summarized in *Supplementary file 1* (New dataset) and *Supplementary file 2* (IMMREP 2022 dataset).

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
