## [Editor Report · eLife assessment]

This study presents a **useful** tool for predicting TCR specificity with **compelling** evidence for improvements over prior art. This work/tool will be broadly relevant to computational biologists and immunologists.

---

## [Referee Report · Reviewer #1 (Public Review)]

In this article, different machine learning models (pan-specific, peptide-specific, pre-trained, and ensemble models) are tested to predict TCR-specificity from a paired-chain peptide-TCR dataset. The data consists of 6,358 positive observations across 26 peptides (as compared to six peptides in NetTCR version 2.1) after several pre-processing steps (filtering and redundancy reduction). For each positive sample, five negative samples were generated by swapping TCRs of a given peptide with TCRs binding to other peptides. The weighted loss function is used to deal with the imbalanced dataset in pan-specific models.

The results demonstrate that the redundant data introduced during training did not lead to performance gain; rather, a decrease in performance was observed for the pan-specific model. The removal of outliers leads to better performance.

To further improve the peptide-specific model performance, an architecture is created to combine pan-specific and peptide-specific models, where the pan-specific model is trained on pan-specfic data while keeping the peptide-specfic part of the model frozen, and the peptide-specific model is trained on a peptide-specific dataset while keeping the pan-specific part of the model frozen. This model surpassed the performance of individual pan-specific and peptide-specific models. Finally, sequence similarity-based predictions of TCRbase are integrated into the pre-trained CNN model, which further improved the model performance (mostly due to the better discrimination of binders and non-binders).

The prediction for unseen peptides is still low in a pan-specific model; however, an improvement in prediction is observed for peptides with high similarity to the ones in the training dataset. Furthermore, it is shown that 15 observations shows satisfactory performance as compared to the ~150 recommended in the literature.

Models are evaluated on the external dataset (IMMREP benchmark). Peptide-specific models performed competitively with the best models in the benchmark. The pre-trained model performed worst, which the authors suggested could be because of positive and negative sample swapping across training and testing sets. To resolve this issue, they applied the redundancy removal technique to the IMMER dataset. The results agreed with earlier conclusion that the pre-trained models surpassed peptide-specific models and the integration of similarity-based methods leads to performance boost. It highlights the need for the creation of a new benchmark without data redundancy or leakage problems.

The manuscript is well written, clear and easy to understand. The data is effectively presented. The results validate the drawn conclusions.

---

## [Referee Report · Reviewer #2 (Public Review)]

Summary:

The authors describe a novel ML approach to predict binding between MHC-bound peptides and T-Cell receptors. Such approaches are particularly useful for predicting the binding of peptide sequences with low similarity when compared to existing data sets. The authors focus on improving dataset quality and optimizing model architecture to achieve a pan-specific predictive model in hopes of achieving a high precision model for novel peptide sequences.

Strengths:

Since assuring the quality of training datasets is the first major step in any ML training project, the extensive human curation and computational analysis and enhancements made in this manuscript represent a major contribution to the field. Moreover, the systematic approach to testing redundancy reduction and data augmentation is exemplary, and will significantly help future research in the field.

The authors also highlight how their model can identify outliers and how that can be used to improve the model around known sequences, which can help the creation and optimization of future datasets for peptide binding.

The new models presented here are novel and built using paired α/β TCR sequence data to predict peptide-specific TCR binding, and have been extensively and rigorously tested.

Weaknesses:

Achieving an accurate pan-specific model is an ambitious goal, and the authors have significant difficulties when trying to achieve non-random performance for prediction of TCR binding to novel peptides. This is the most challenging task for this kind of model, but also the most desirable when applying such models to biotechnological and bioengineering projects.

The manuscript is a highly technical and extremely detailed computational work, which can make the achievements and impact of the work hard to parse for application-oriented researchers, and still hard to translate to real-world use-cases for TCR specificity predictions.

---

## [Author Response]

The following is the authors’ response to the original reviews.

We thank the editor and reviewers for their valuable feedback and comments. Below we have addressed all points carefully and have, when needed, revised the manuscript accordingly.

Note that we have taken the opportunity to correct minor typos and unclear text in the revised manuscript.

Of importance to the editors and reviewers, we detected a few minor factual errors in the method section, which we have now corrected. The first error was that we wrongfully stated that our final dataset had 6358 unique TCRs, whereas it was in fact 6353 unique TCRs. The second error was that we stated that the maximum length of CDR1ꞵ was 5, where it was in fact 6. The last error was that we stated that we used a Levenshtein distance of at least 3 to discard similar peptides when swapping the TCRs to generate negatives. This should have been a Levenshtein greater than 3, to match the script we used to generate negatives (though no peptides had a Levenshtein distance of exactly 3).

**eLife assessment**
This important study reports on an improved deep-learning-based method for predicting TCR specificity. The evidence supporting the overall method is compelling, although the inclusion of real-world applications and clear comparisons with the previous version would have further strengthened the study. This work will be of broad interest to immunologists and computational biologists.

It is not fully clear to us what is meant by “clear comparisons with the previous version”. In the manuscript we consistently compare the performance of each novel approach introduced to that of the ancestor NetTCR-2.1. Further, we concluded the manuscript with a performance to a large set of current state-of-the-art methods by training and evaluating the novel modeling framework on the IMMREP22 benchmark data.

We agree that the manuscript can be improved by including a brief discussion of real-life applications of models for prediction of TCR specificity, and have included a brief text in the introduction.

**Reviewer #1 (Recommendations For The Authors):**
It was a great pleasure to read this article. All the concepts and motivations are clearly defined. I have just a few questions.What was the motivation behind employing a 1:5 positive-negative ratio? Could it be the cause of worse performance in the case of outliers?

The ratio 1:5 is based on results from earlier work [36561755]. In this work, negatives were constructed as a mix of swapped and true (i.e measured) negatives with a ratio 1:5 for each. This work demonstrated a slight gain when including both types of negatives compared to only using swapped. In a subsequent publication [https://doi.org/10.1016/j.immuno.2023.100024], it demonstrated that optimal performance was obtained when only including swapped negatives (again in a ratio 1:5). Given this, we maintained this approach in the current work. It is clear that this choice is somewhat arbitrary, and that further work is needed to fully address this issue and the general issue of how to best generate negatives for ML of TCR specificity. Such work is in our view however beyond the scope of the current manuscript.

Why is the patience of 200 epochs for peptide-specific models and 100 epochs for pan-specific and pre-trained models used in the context of the early stopping mechanism?

We observed that the loss curve was overall very stable in the case of pan-specific training, likely due to the large amount of data included in this training. Therefore, these models were less likely to become stuck in a local minimum during training, meaning that a lower patience for early stopping would not prevent the model from learning optimally. In contrast, we found for some peptides that the loss curve was very erratic, and would sometimes become stuck in a local minimum for an extended time. To resolve this, the patience was increased from 100 to 200, which resulted in a better chance to escape these minima, as well as a better overall performance.

Why is weight 3.8 used in the weighted loss function in the pan-specific model?

The weighted loss was scaled with a division factor (c) of 3.8, in order to get an overall loss that was comparable to training without sample weights. This was primarily done to better compare the two approaches (scaling and no scaling) in terms of loss, and not so much to improve the training itself, as we already use a relatively conservative sample weight scaling based on log2. We have added a brief sentence to clarify this in the manuscript.

**Reviewer #2 (Recommendations For The Authors):**
This work is the evolution of previous studies that developed the NetTCR platform, and in a previous paper cited in this study, the authors explore the paired dataset approach with "paired α/β TCR sequence data". In this manuscript, the authors should make clear what advances were made when compared to the previous study. This is not clear, although extensive reference is made to NetTCR 2.0 and 2.1. Differences are scattered throughout the manuscript, so I would suggest a section or paragraph clearly delineating the advances in model architecture and training when compared to previous versions recently published.

It is not clear to us when the reviewer is referring to when stating “the authors should make clear what advances were made when compared to the previous study”. Throughout the manuscript we consistently compare the performance of each novel approach introduced to that of the ancestor NetTCR-2.1. In addition, we briefly discuss all of the changes to the architecture and training at the start of the discussion section. Further, we concluded the manuscript with a performance to a large set of current state-of-the-art methods by training and evaluating the novel modeling framework on the IMMREP22 benchmark data. It is correct that the advances are described progressively by introducing each novel approach one by one, i.e. refining the machine learning model architecture and training setup, data denoising in terms of outlier identification in the training data, new model architectures combining the properties of a pan- and peptide-specific model, and integration of similarity based approach to boost model performance. We believe this helps better justify the relevance of each of the novel approaches introduced.

In Figure 3, the colors have labels, but they are not explained in the legend or in the text. This makes it very difficult to understand the data in the various columns. Also, since it represents the Mean AUC, the data would be best displayed with a boxplot or a mean and bars for variance.

We agree, and have changed Figure 3 and its corresponding AUC 0.1 figure (Supplementary Figure 1) into a boxplot. We also further clarified what the different models were in the figure text.

Given the potential impact of this work on bioengineering and biotechnology, I would suggest adding a paragraph or section to the discussion where potential applications of the current model, or examples of applications of previous (or competing) models have been used to further biological research.

We agree and have added a brief sentence in the introduction to outline biotechnological applications of models for prediction of TCR specificity.